

# Novel ACE2 protein interactions relevant to COVID-19 predicted by evolutionary rate correlations

Austin A. Varela*, Sammy Cheng and John H. Werren*

Department of Biology, University of Rochester, Rochester, New York, United States
* These authors contributed equally to this work.

Corresponding author
John H. Werren,
jack.werren@rochester.edu

## ABSTRACT

Angiotensin-converting enzyme 2 (ACE2) is the cell receptor that the coronavirus SARS-CoV-2 binds to and uses to enter and infect human cells. COVID-19, the pandemic disease caused by the coronavirus, involves diverse pathologies beyond those of a respiratory disease, including micro-thrombosis (micro-clotting), cytokine storms, and inflammatory responses affecting many organ systems. Longer-term chronic illness can persist for many months, often well after the pathogen is no longer detected. A better understanding of the proteins that ACE2 interacts with can reveal information relevant to these disease manifestations and possible avenues for treatment. We have undertaken an approach to predict candidate ACE2 interacting proteins which uses evolutionary inference to identify a set of mammalian proteins that "coevolve" with ACE2. The approach, called evolutionary rate correlation (ERC), detects proteins that show highly correlated evolutionary rates during mammalian evolution. Such proteins are candidates for biological interactions with the ACE2 receptor. The approach has uncovered a number of key ACE2 protein interactions of potential relevance to COVID-19 pathologies. Some proteins have previously been reported to be associated with severe COVID-19, but are not currently known to interact with ACE2, while additional predicted novel ACE2 interactors are of potential relevance to the disease. Using reciprocal rankings of protein ERCs, we have identified strongly interconnected ACE2 associated protein networks relevant to COVID-19 pathologies. ACE2 has clear connections to coagulation pathway proteins, such as Coagulation Factor V and fibrinogen components FGA, FGB, and FGG, the latter possibly mediated through ACE2 connections to Clusterin (which clears misfolded extracellular proteins) and GPR141 (whose functions are relatively unknown). ACE2 also connects to proteins involved in cytokine signaling and immune response (*e.g.* XCR1, IFNAR2 and TLR8), and to Androgen Receptor (AR). The ERC prescreening approach has elucidated possible functions for relatively uncharacterized proteins and possible new functions for well-characterized ones. Suggestions are made for the validation of ERC-predicted ACE2 protein interactions. We propose that ACE2 has novel protein interactions that are disrupted during SARS-CoV-2 infection, contributing to the spectrum of COVID-19 pathologies.

## INTRODUCTION

The coronavirus SARS-CoV-2 is causing severe pathologies and death among infected individuals across the planet. In addition to the symptoms expected from a respiratory disease, the infection can develop systemic manifestations (*Gupta et al., 2020*; *Terpos et al., 2020*; *Siddiqi, Libby & Ridker, 2021*). As a consequence, a wide range of pathologies are associated with COVID-19, including vascular system disruption, the extensive formation of blood clots (thrombosis) resulting in microvascular injury and stroke (*Magro et al., 2020*; *Connors & Levy, 2020*), gastrointestinal complications (*Luo, Zhang & Xu, 2020*) cardiac and kidney pathologies, ocular and dermatological symptoms (*Bouaziz et al., 2020*), neurological manifestations (*Niazkar et al., 2020*; *Taquet et al., 2021*), male infertility (*Khalili et al., 2020*), and a Kawasaki-like blood and heart disorder in children (*Jones et al., 2020*; *Morand, Urbina & Fabre, 2020*). A severe and often lethal immunoreaction can occur from respiratory and other infection sites, termed a "cytokine storm" (*Chen et al., 2020*). Even after acute SARS-CoV-2 infection has passed, individuals can suffer a suite of complications for many months, termed "Long Haul" syndrome (*López-León et al., 2021*), and the causes of these syndromes are not well understood.

Angiotensin-converting enzyme 2 (ACE2) is of obvious interest because it is a primary receptor for SARS-CoV-2 entry into human cells (*Lan et al., 2020*). However, ACE2 also plays a role in other important processes, such as regulation of blood pressure and vasodilation by the renin-angiotensin system (RAS), and protein digestion in the gut (*Kuba et al., 2010*). SARS-CoV-2 binding to ACE2 leads to a downregulation in ACE2 function (*Verdecchia et al., 2020*) which may be linked to the systemic damage by COVID-19 (*Medina-Enríquez et al., 2020*). It has been proposed that ACE2 receptor degradation during SARS-CoV-2 infection disrupts protection by ACE2 from inflammatory processes through the RAS and bradykinin pathways, possibly explaining patterns of COVID-19 severity with age and sex (*Bastolla, 2020*; *Bastolla et al., 2021*). As well as being a cell receptor, a circulating soluble form of the ectodomain of ACE2 (sACE2) is shed from cells and found in blood plasma, but the biological function of circulating ACE2 remains relatively unknown. Elevated levels of sACE2 have been detected in critically ill COVID-19 patients (*Van Lier et al., 2021*) which coincides with a reduced expression of membrane-bound ACE2 (*Medina-Enríquez et al., 2020*), and a recent study indicates that sACE2 may assist SARS-CoV-2 entry into cells *via* other receptors (*Yeung et al., 2021*).

In general, ACE2's protein-protein interaction network is likely to contribute to COVID-19 pathologies, due to ACE2's role in systemic processes that are disrupted by the infection. Therefore, a fuller knowledge of ACE2 protein interactions is important to a better understanding of COVID-19 pathologies, including those that go beyond respiratory illness.

Common methods to identify protein-protein interactions include protein co-localization and precipitation, genetic manipulation, and proteomic profiling (*Rao et al., 2014*). Evolutionary approaches have also been used to evaluate protein interactions (*De Juan, Pazos & Valencia, 2013*), particularly to identify functional domains within proteins based on sequence conservation in evolution. Another set of methods utilize

evolutionary rate correlations (also called evolutionary rate covariance or evolutionary rate coevolution). The concept is that coevolving proteins will show correlated rates of change across evolution (*Wolfe & Clark, 2015*). The approach has been used to detect physical interactions within and among proteins, as well as shared functionality not involving physical interaction, such as within metabolic pathways (*Clark, Alani & Aquadro, 2012*). For example, it has been employed to identify gene networks for post-mating response (*Findlay et al., 2014*), ubiquitination (*Böhm et al., 2016*), and recombination (*Godin et al., 2015*), and more recently to identify DNA repair genes (*Brunette et al., 2019*), cadherin-associated proteins (*Raza et al., 2019*), mitochondrial-nuclear interactions (*Yan, Ye & Werren, 2019*), and a mitochondrial associated zinc transporter (*Kowalczyk et al., 2021*), with subsequent experimental support. Evolutionary rate correlation (ERC) approaches are relatively inexpensive screening tools for detecting candidate protein interactions, and can also detect novel protein interactions that are not readily found in more traditional proteomic and genetic approaches (*Colgren & Nichols, 2019*; *Yan, Ye & Werren, 2019*). As such, "the ERC method should be a part of the toolkit of any experimental cell or developmental biologist" (*Colgren & Nichols, 2019*).

We have developed an evolutionary rate correlation (ERC) method that uses well-established phylogenies based on multiple lines of evidence (*e.g. Misof et al., 2014* for insects and *Kumar et al., 2017* for mammals) and calculates protein evolutionary rates for terminal branches for different proteins across a set of related species (Fig. 1). The approach is predicated on the idea that proteins that have strong evolutionary rate correlations are more likely to have functional interactions that are maintained by their coevolution, a conclusion supported by its predictive power in identifying known nuclear-mitochondrial encoded protein interactions in insects (*Yan, Ye & Werren, 2019*). That study also found that nuclear-encoded proteins and amino acids in contact with their mitochondrial-encoded components (*e.g.* oxidative phosphorylation proteins or mitochondrial ribosomal RNA) have significantly stronger ERCs than those not directly in contact. This result implicates physical interactions between proteins as one driver of evolutionary rate correlations, at least among nuclear-mitochondrial interactions. Other studies have found evolutionary rate correlations among proteins that do not make direct contact, such as in metabolic pathways (*Clark, Alani & Aquadro, 2012*).

We have developed a reciprocal rank approach to identify ACE2 associated networks and propose that these strongly coevolving proteins reveal ACE2 protein interactions that could be disrupted by COVID-19, thus contributing to its diverse pathologies. Particularly noteworthy are strong connections to coagulation pathway proteins, cytokine signaling, inflammation, immunity, and viral disease response.

It is important to note that our approach cannot be used to study coronavirus-ACE2 coevolution. The reason is that coronaviruses move between mammalian species and therefore do not have the same phylogenetic history as mammalian proteins, a prerequisite for the approach. We are also not asserting that coronavirus pressure is causing the evolution of ACE2 observed in the ERCs. Rather, it is our proposition that the ACE2 ERCs are revealing evolved mammalian protein interactions that are not caused by, but could be

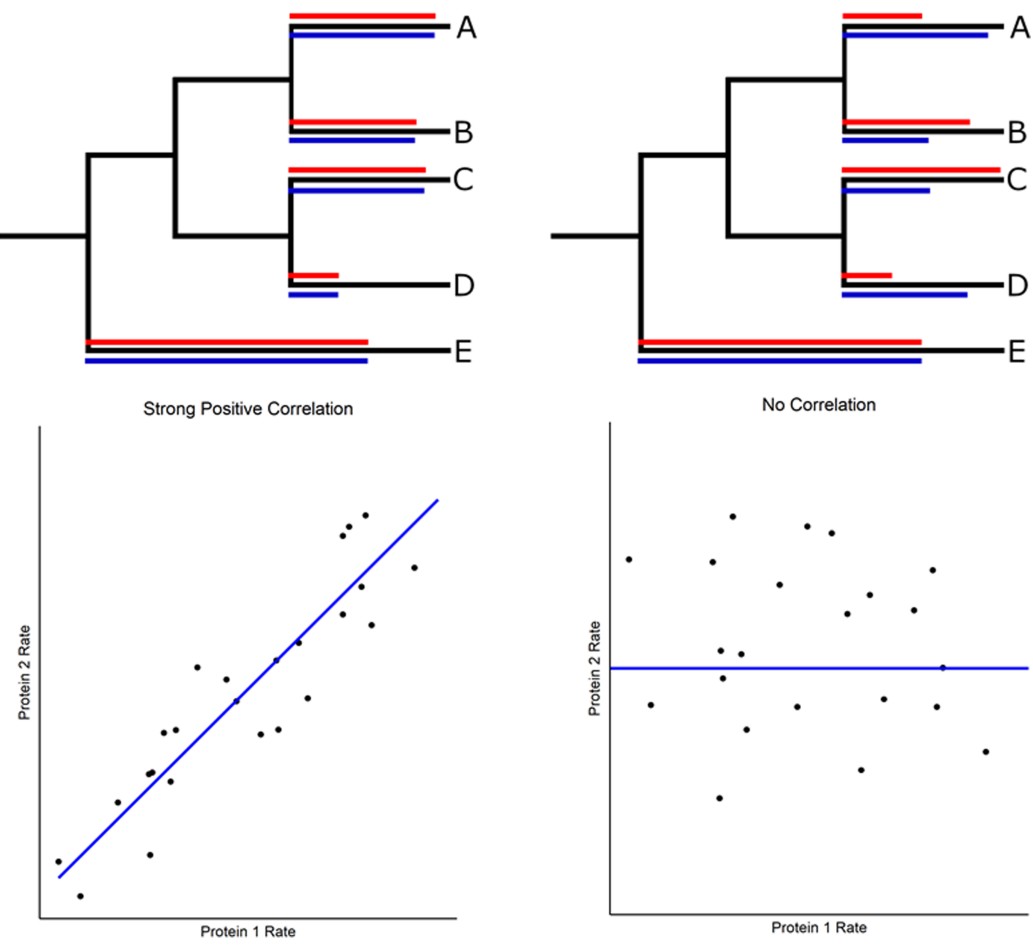

**Figure 1 Evolutionary rate correlations.** The Spearman rank correlations between two proteins are calculated based on rates of protein evolution on terminal branches of a phylogeny. The relative rates of two proteins (red and blue lines) are shown in the hypothetical phylogenetic trees. Correlated and uncorrelated protein rates are illustrated below using a larger number of terminal branches (data points) than presented in the phylogeny.

relevant to COVID-19 pathologies, due to disruption of pathways in which ACE2 is normally involved.

We recognize that the predicted protein interactions detected by the ERC approach may not be causal for COVID-19 pathologies. However, the ERC connections to coagulation pathways, cytokine signaling, and immunity are striking and suggest a possible role of these ACE2 protein interactions in COVID-19 syndromes. The ERC results may also have implications for ACE2's role in the regulation of vasodilation through the renin-angiotensin system (RAS), cardiovascular disease, and protein digestion and absorption in the gut (*Kuba et al., 2010*). Furthermore, the ERC analysis reveals possible novel connections for coagulation pathway and cytokine-signaling proteins that may be worthy of further investigation. Therefore, validation studies of the ERC predictions described here are desirable, both for possible applications to COVID-19 disease and treatment, and for investigations of other important biological processes.

## MATERIALS & METHODS

### Taxon selection and data collection

Our evolutionary rate correlation (ERC) approach requires orthologous protein sequence data across a large number of taxa with well-defined phylogenetic relationships. Calculation of evolutionary rates requires a resolved phylogeny of the taxa analyzed that is scaled to evolutionary time. Our ERC calculations utilize the TimeTree (*Kumar et al., 2017*) to generate a time-scaled phylogenetic tree using the mammalian taxa that are represented in OrthoDB sequence data (Fig. S1). The tree generated is in units of millions of years and is based on a compilation of many phylogenetic-dating related studies. The tree is utilized as a base topology in phylogenetic analysis and its branch lengths are used to measure time for calculating evolutionary rates from the resultant individual protein trees (Fig. 1). Additional details on the data set are provided in the Supplemental Text. Large files are deposited in FigShare (https://doi.org/10.6084/m9.figshare.14637450) and are listed in Table S1.

Well-defined orthologous sequence data is sourced from OrthoDB (*Kriventseva et al., 2019*) at the "mammalia" (taxonomic id: 40674) taxonomic level. Since OrthoDB sequence data is gathered from a variety of sources and clustered algorithmically (unsupervised), primarily based on sequence similarity (*Kriventseva et al., 2015*), related paralogous proteins are often clustered with each other even if canonically annotated as functionally distinct proteins (Table S2). Additionally, since the data sources for sequences can have varying levels of completeness, most ortholog groups on OrthoDB are missing sequence data for one or more taxa represented in the database. So, a majority of the data we initially selected was from single-copy ortholog groups with at least 90 of the 108 possible mammalian taxa present. In addition, some proteins with a possibly relevant function to COVID-19 pathologies (such as XCR1, and IFNAR2) or other relevant pathways in ortholog groups that did not meet the initial criteria, but that had minimal paralogy issues, were included. Paralogous sequences were manually disambiguated based on published protein annotations and phylogenetic analysis. If a taxon in a given sequence had duplicate sequences that could not be disambiguated, the taxon was excluded in phylogenetic and ERC calculations for the specific proteins involved. In total, 1,953 orthologous protein groups are used in analyses.

### Phylogenetic calculations and protein alignments

To prepare orthologous sequence data for ERC calculation, each set of protein sequences are first aligned using the MAFFT software package (*Katoh & Standley, 2013*) using the following arguments: "−−maxiterate 1000 −−localpair −−anysymbol". Since the sequences come from data sources with varying levels of quality and multiple alignment programs can be imperfect, the aligned sequences must then be trimmed. The alignments are trimmed using the trimAl software package (*Capella-Gutiérrez, Silla-Martínez & Gabaldón, 2009*) using the "-automated1" argument to remove poorly aligned regions. These final prepared alignments are then used to generate maximum-likelihood phylogenies. The IQ-TREE software package (*Minh et al., 2020*) is used to estimate

protein branch lengths (equivalent to average substitution counts per site). Specifically, the "LG+F+G+I" model (which utilizes an empirically derived amino acid substitution matrix) is used with the following additional parameters: "-B 1000 -st AA -seed 1234567890" and the TimeTree phylogeny is provided to constrain output tree topology to reduce possible branch length estimation errors with the "-g" option. These trees are the basis of ERC calculations. Protein branch lengths are based on the average number of changes in amino acids at each residue in the alignment. The resultant branch lengths are paired with corresponding branches in the TimeTree to quantify branch-specific rates to be used for ERC calculations (described below). ERCs calculated with the more complete phylogeny (108 species) had short branch problems in oversampled taxonomic groups (described below and in the Supplemental Text). We therefore used a reduced phylogeny consisting of 60 taxa for subsequent ERC analyses.

## Calculation of ERCs

Our evolutionary rate correlation (ERC) method is designed to predict protein-protein interactions using evolutionary data (*Yan, Ye & Werren, 2019*), and is based on protein evolutionary rates on terminal branches of the mammalian phylogeny (Fig. 1). We found that the more complete phylogeny (108 species) had short branch problems that inflate ERC spearman rank correlations (discussed in Supplemental Text). Most notably, there was an association between branch time and protein rate for many proteins, with oversampling in some taxonomic units (*e.g.* in Primates and Rodentia) leading to many ERCs being driven by relatively short branches (Supplemental Text). We attempted to control for these effects initially by using a partial correlation method, but found that it was not sufficient due to correlations between residuals and branch time (Supplemental Text). We then removed taxa that contributed short branches in our phylogeny based on either a 20MY or 30MY divergence time threshold (Supplemental Text) and recalculated branch rates for all proteins. We found that the 30MY threshold short branch removal eliminated significant branch time to protein rate correlations for the majority of proteins (87.5%). The resultant rate data no longer has branch time to branch rate as a confounding cofactor, and the ERCs themselves are no longer biased by extremely short branches and taxonomic oversampling (Supplemental Text). The resulting data set is composed of 60 taxa and is used in our subsequent analyses of ERCs.

Using the adjusted data set, ERCs are calculated for every possible combination of protein pairs for which a tree has been generated. Every protein pair for which an ERC is calculated has each respective tree and the TimeTree topology is pruned to only include the shared taxa between the two, using the "ETE3" Python package (*Huerta-Cepas, Serra & Bork, 2016*). ETE3 is also used to extract the terminal branch lengths of each pruned tree. Evolutionary rates are calculated by dividing the terminal protein-tree branch lengths (average substitutions per site) by the corresponding branch in the TimeTree (measured in millions of years). Terminal branches are used for calculations as they do not have shared evolutionary histories and are therefore independent. The resulting rates have the unit of average substitution per site per millions of years. Given the resultant rates, evolutionary

rate correlations are then calculated by performing a Spearman's rank correlation test (*Yan, Ye & Werren, 2019*) using the Python package "SciPy" (*Virtanen et al., 2020*).

## Multiple test corrections

*P*-values are corrected using the Benjamini–Hochberg FDR multiple-test correction procedure implemented in the Python package "statsmodels" (*Seabold & Perktold, 2010*). The FDR correction is applied to each respective protein's set of ERCs. Correlation test results are non-directional, but FDR corrections are dependent on the rank of each correlation's *p*-values. Since the rank of each correlation test value on respective protein lists vary, the FDR-corrected *p*-values of a given protein pair can differ depending on the specific ERC set. An ERC is considered significant if the FDR-corrected *p*-value is less than 0.05.

## ERC set enrichment analysis

To summarize the common biological function of proteins that tend to have strong ERCs, gene set enrichment analysis is performed on the top 2% of ERCs (by ρ) of each protein(s) of interest (including the protein itself), including only proteins with ERCs that are significant following an FDR correction at a significance level of 0.05. At most, a protein of interest will have 41 proteins included for enrichment analysis (2% of the total 1,953 proteins plus itself). Protein set enrichment analyses are performed using the Enrichr service (*Xie et al., 2021*) *via* the Python bindings provided by the "GSEApy" Python package (*Fang et al., 2021*) given the background of the full set of 1,953 proteins. We calculate enrichment results for ACE2 and all of its top 20 ERC partners. Additional enrichment analyses were also performed on a case-by-case basis based on relevance, including the reciprocal rank networks. Enrichments are performed using selected relevant term databases: KEGG_2019_Human, GO_Biological_Process_2018, GO_Cellular_Component_2018, GO_Molecular_Function_2018, Reactome_2016, WikiPathways_2019_Human, Tissue_Protein_Expr_from_Human_Proteome_Map, Tissue_Protein_Expr_from_ProteomicsDB, and Jensen_TISSUES.

Enrichment results for terms that are significant at FDR-adjusted *p* < 0.05 for all analyses are placed into a single table, organized by the enrichment term database (File 3). The outputs from different databases can contain redundant terms to each other, so only the most significant of the redundant terms are reported for any enrichment analysis in the main text.

## Reciprocal rank network (RRN) generation

To evaluate and visualize the strongest ERCs centered around proteins of interest, "reciprocal rank networks" (RRNs) are produced. Reciprocal ranks refer to the fact that a significant ERC between two proteins can have different ranks in the two respective protein ERC lists because some proteins have more and higher ERCs than others. To focus on networks of proteins with strong reciprocal rank correlations, we have constructed networks based on proteins with reciprocal ranks of 20 or less (RR20), which is the top 1% in each protein's highest ERCs based on ρ values. Specifically, we have developed an ACE2

centric reciprocal rank network by the following steps (1) for ACE2, select its top 1% (20) proteins, (2) for each of those proteins, select additional proteins in their ERC list with reciprocal rank 20 or less, and then (3) Given the core set of proteins generated in the previous two steps, connect proteins which have a unidirectional rank of 20 or less.

The resultant network represents the strongest ERCs centered around a protein of interest (in this case ACE2), along with the immediate neighborhood of the strongest ERCs surrounding the protein of interest. The ACE2 Core Reciprocal Rank (CRR) was initiated with the four proteins to which ACE2 has RR20 ERCs (CLU, TMEM63C, FAM3D, and L1CAM), with GPR141 added due to its RR1 strong connection to CLU and unidirectional connection to ACE2. ACE2 also has highly significant ERCs to proteins that do not rank it in their top one percent. Therefore, a similar approach has been used to generate an ACE2 reciprocal network initiated with the top 10 proteins to which ACE2 has highly significant ERCs, but are not reciprocally RR20 ranked, with a subsequent one cycle RR20 built upon these. This ACE2 Unidirectional Reciprocal Rank Network (URR) contains strong network connections to ACE2 through its high unidirectional ERCs. Steps 2–3 were omitted as the network becomes extremely large following just the first step, and our focus is on examining close connections to ACE2 based on ERC analysis.

## ERCs within and between protein complexes

To compare whether calculated ERCs are stronger between known interactions *versus* non-interactions, the protein complex database, CORUM (*Giurgiu et al., 2019*), was used to retrieve known complexes. The "Core Complex" dataset was downloaded and filtered for human complexes to eliminate redundancy, resulting in 233 protein complexes from this CORUM data set which have two or more components present in our 1,953 protein ERC set, representing 258 pairwise ERC comparisons. As these protein complexes have redundancy (*i.e.* some complexes contain overlapping protein pairs), the set was further restricted to complexes containing unique protein components—resulting in 139 effective unique complexes considered. To test whether ERCs within complexes are higher than between complexes, all pairwise ERCs within complexes were compared to the median $\rho$ value for each pair to proteins present in non-redundant CORUM set that are not in complex with either of these proteins. A Wilcoxon matched signed-rank test was performed using the "wilcox.test" function in base R (version 3.6.1; with parameters "paired" and "exact" set to "TRUE") on the in-complex $\rho$ values and the median out-of-complex $\rho$ values, to test if the in-complex $\rho$ values were significantly greater than the median out-of-complex $\rho$ values. In addition, as there were many complexes with a majority of subcomponents not present in our 1,953 datasets, the likelihood of individual pairs directly interacting within the complex decreases with the increasing number of proteins in a complex. Therefore, an additional Wilcoxon matched signed-rank test was performed on members of protein complexes composed of five or fewer proteins.

## Testing for taxonomic effects

We use three methods to test for taxonomic effects on the calculated ERCs, (1) multiple linear regression, (2) analysis of covariance (ANCOVA), and (3) non-parametric

independent contrasts (*Garland, Harvey & Ives, 1992*). For the regression and ANCOVA approaches, rate data was grouped by mammalian taxonomic orders accessed *via* ETE3 (*Huerta-Cepas, Serra & Bork, 2016*) and treated as an independent variable.

The independent contrasts test uses the mammalian topology previously created with TimeTree (*Kumar et al., 2017*) to generate independent contrasts within the phylogeny. Statistical tests for each method are performed using base R (version 3.6.1). See the Supplemental Text for details.

## Testing whether branch rates increase when extending branch time within clades

To test whether increasing branch length results in increasing protein evolutionary rate, we selected separate phylogenetic groups (clades) from the full phylogeny (Fig. S1) that contain short branch lengths. Protein evolutionary rate was calculated for each protein on the short branch, and then sequentially recalculated after removing adjacent taxa to extend the branch internally (Fig. S5). In this way, the protein evolutionary rate was examined as branches are extended internally in independent clades within the tree. Comparing original branches to the 20MY correction resulted in 12 clades for which time scales change between 20MY and 30MY corrections, and 16 clades for which time scales change between 0MY and 30MY. Tests on each branch's rate against the respective adjusted rate were performed using two-tailed Wilcoxon Matched Signed Rank Tests (Base R v3.6.1), first for proteins of interest (*e.g.* ACE2) and then for the full protein set. Results are described in the Supplemental Text.

# RESULTS

## Basic approach

The basic methods are outlined here to provide context for the results which follow. To identify candidate protein interactions using evolutionary rate correlation, we utilized the consensus TimeTree phylogenetic reconstruction for mammalian species (*Kumar et al., 2017*). A total of 1,953 proteins (including ACE2) were aligned and evolutionary rates for each protein were then calculated for terminal branches of the tree (Fig. 1). This was determined by dividing the protein-specific branch length on each terminal branch by terminal branch time from the consensus tree (*Yan, Ye & Werren, 2019*). Maximum likelihood branch lengths were estimated in IQ-TREE (*Minh et al., 2020*) using an empirical amino acid substitution matrix (see methods for details). To investigate evolutionary rate correlations (ERCs) among proteins, Spearman rank correlations were calculated for every protein pair using terminal branch rates (Fig. 1). Due to the large number of comparisons, a Benjamini–Hochberg false discovery rate (FDR) correction was calculated for each protein's ERC set (significance threshold $\alpha = 0.05$). We subsequently found that many proteins show a positive correlation between terminal branch time and evolutionary rate, and observed that short branches in relatively oversampled taxa significantly contributed to this correlation (Supplemental Text). We, therefore, removed species that accounted for short branches, which eliminated the protein evolutionary rate

to branch time correlation (see Methods and Supplemental Text for details). ERCs were then recalculated, and our ERC analyses are based on this set of 60 taxa.

In addition, we tested whether the observed lower rates of protein evolution for short terminal branches in the phylogeny are due to rates actually increasing over evolutionary time, *versus* a taxonomic effect. This was accomplished by examining changes in protein rates in independent clades as terminal branches were effectively extended by selective removal of flanking taxa. The analysis shows that evolutionary rates for many proteins increase as branch length is increased (described in more detail in Supplemental Text, Fig. S4). A possible explanation for the pattern is that protein coevolution is mostly episodic, and short branches in a phylogeny are less likely to capture such events. In additional analyses, we tested for but did not find significant confounding effects of taxonomy on the ERC results (Supplemental Text).

Our analyses are focused on candidate protein interactions involving ACE2 using evidence of highly significant ERCs. For this purpose, we first examine proteins in ACE2's highest 2% of ERCs (top 40 proteins), all of which are highly significant after FDR correction (Table 1). Some of these ACE2 ERC proteins have been previously implicated in severe COVID-19 or SARS-CoV-2 gene expression effects on infected cells. However, while they have not been previously identified as having protein interactions with ACE2, this is predicted by our ERC analysis.

X-C Motif Chemokine Receptor 1 (XCR1) provides an illustrative example. XCR1 is a cytokine signaling receptor and ACE2's 2nd highest ranked ERC, with a highly significant evolutionary rate correlation. XCR1 is in a small genomic region that is implicated in severe COVID-19 by genome-wide association studies (*Severe Covid-19 GWAS Group, 2020*; *Fricke-Galindo & Falfán-Valencia, 2021*). Another example is Interferon alpha/beta receptor 2 (IFNAR2) which, in a genome-wide association study (GWAS) and multi-omic analysis by *Pairo-Castineira et al. (2021)*, was implicated in severe COVID-19. We therefore added it to our analysis, and surprisingly found it to be highly ranked (5th) among ACE2 ERCs. Clusterin (CLU) is the 3rd strongest ERC of ACE2 and the ACE2-CLU pair show high reciprocal ranks to each other (3rd in ACE2's set, 8th in CLU's set). CLU prevents the aggregation of misfolded proteins in the blood and delivers them to cells for degradation in lysosomes (*Sánchez-Martín & Komatsu, 2020*). CLU connects to key proteins in the coagulation pathway based on its reciprocal rank network ("ERC Reciprocal Rank Networks Implicate Coagulation Pathways and Immunity", Fig. 2). CLU has been implicated in coronavirus infections, as one of only two proteins showing significant expression changes in cells infected by three different coronaviruses tested, including SARS-CoV-2 (*Singh et al., 2021*). The examples above lend credence to the proposition that the ERC approach is detecting ACE2 protein interactions that have implications to COVID-19.

Differences in ERC rank between protein pairs for the same correlation can occur because some proteins have higher and more extensive ERC connections than others. As a result, while two proteins can have a significant ERC with each other, each one's rank may differ in their respective ERC lists, as illustrated for ACE2 and GEN1 (Table 1). GEN1 (Flap endonuclease GEN homolog 1) is ACE2's top-ranked ERC, and is a DNA nuclease

**Table 1 Top 2% ERCs for ACE2 and GEN1.**

| Protein | ACE2 Rank | ACE2's Partner Rank | ρ | P | FDR | Protein | GEN1 Rank | GEN1's Partner Rank | ρ | P | FDR |
|---|---|---|---|---|---|---|---|---|---|---|---|
| GEN1 | 1 | 203 | 0.67 | 4.3E−08 | 4.2E−05 | IFNLR1** | 1 | 1 | 0.89 | 3.2E−20 | 6.2E−17 |
| XCR1 | 2 | 37 | 0.67 | 3.2E−08 | 4.2E−05 | CC2D1B** | 2 | 1 | 0.84 | 5.3E−16 | 5.2E−13 |
| CLU** | 3 | 8 | 0.63 | 3.1E−07 | 1.5E−04 | MUC15** | 3 | 15 | 0.84 | 4.2E−15 | 2.7E−12 |
| TMEM63C** | 4 | 11 | 0.63 | 2.0E−07 | 1.3E−04 | SPZ1 | 4 | 30 | 0.82 | 5.0E−14 | 1.4E−11 |
| IFNAR2 | 5 | 392 | 0.62 | 2.5E−06 | 6.1E−04 | SLC10A6** | 5 | 2 | 0.82 | 1.2E−14 | 5.9E−12 |
| KIF3B | 6 | 26 | 0.60 | 1.7E−06 | 4.9E−04 | ARID4A** | 6 | 9 | 0.81 | 2.0E−14 | 8.0E−12 |
| ITPRIPL2 | 7 | 364 | 0.59 | 1.7E−06 | 4.9E−04 | RAD51AP2 | 7 | 22 | 0.81 | 6.7E−14 | 1.6E−11 |
| FAM227A | 8 | 175 | 0.59 | 1.8E−06 | 4.9E−04 | TESPA1** | 8 | 2 | 0.81 | 3.9E−14 | 1.3E−11 |
| TLR8 | 9 | 243 | 0.58 | 3.7E−06 | 7.2E−04 | IFNAR2** | 9 | 9 | 0.80 | 3.4E−12 | 2.6E−10 |
| COL4A4 | 10 | 541 | 0.58 | 3.7E−06 | 7.2E−04 | BCL6B** | 10 | 1 | 0.80 | 1.6E−13 | 3.6E−11 |
| FAM3D** | 11 | 2 | 0.57 | 5.8E−06 | 8.4E−04 | RTL9 | 11 | 54 | 0.80 | 8.0E−13 | 1.1E−10 |
| F5 | 12 | 642 | 0.57 | 4.1E−06 | 7.2E−04 | COL4A5** | 12 | 8 | 0.80 | 4.9E−13 | 8.7E−11 |
| AR | 13 | 22 | 0.57 | 7.7E−06 | 8.8E−04 | APOBR | 13 | 72 | 0.80 | 1.2E−12 | 1.3E−10 |
| TSGA13 | 14 | 423 | 0.57 | 7.1E−06 | 8.8E−04 | COL4A6** | 14 | 19 | 0.79 | 1.6E−12 | 1.6E−10 |
| PLA2G7 | 15 | 387 | 0.57 | 6.0E−06 | 8.4E−04 | TRADD** | 15 | 6 | 0.79 | 6.8E−13 | 1.0E−10 |
| MMS19 | 16 | 387 | 0.56 | 5.9E−06 | 8.4E−04 | FANCG | 16 | 69 | 0.79 | 4.2E−13 | 8.2E−11 |
| AMOT | 17 | 124 | 0.56 | 8.1E−06 | 8.8E−04 | CD180 | 17 | 27 | 0.78 | 8.4E−13 | 1.1E−10 |
| L1CAM** | 18 | 14 | 0.56 | 8.6E−06 | 8.8E−04 | TNFSF18** | 18 | 7 | 0.78 | 2.6E−12 | 2.2E−10 |
| PDYN | 19 | 428 | 0.56 | 7.3E−06 | 8.8E−04 | APOB** | 19 | 1 | 0.78 | 6.7E−13 | 1.0E−10 |
| IQCD | 20 | 158 | 0.56 | 9.2E−06 | 8.9E−04 | MKKS** | 20 | 20 | 0.78 | 8.7E−13 | 1.1E−10 |
| SERPINA5 | 21 | 468 | 0.56 | 2.2E−05 | 1.4E−03 | PIGV | 21 | 8 | 0.78 | 1.6E−12 | 1.6E−10 |
| CERS4 | 22 | 67 | 0.55 | 2.9E−05 | 1.5E−03 | CCDC17 | 22 | 30 | 0.78 | 1.2E−12 | 1.3E−10 |
| CC2D1B | 23 | 467 | 0.55 | 1.1E−05 | 1.0E−03 | DYTN | 23 | 42 | 0.78 | 8.3E−12 | 5.1E−10 |
| GPR141 | 24 | 17 | 0.55 | 1.5E−05 | 1.2E−03 | GNPTAB | 24 | 36 | 0.77 | 1.7E−12 | 1.6E−10 |
| FSCB | 25 | 817 | 0.55 | 2.8E−05 | 1.5E−03 | MTMR11 | 25 | 13 | 0.77 | 2.9E−12 | 2.3E−10 |
| RGR | 26 | 167 | 0.55 | 3.0E−05 | 1.5E−03 | TNFRSF1A | 26 | 25 | 0.77 | 2.0E−12 | 1.7E−10 |
| COL4A5 | 27 | 529 | 0.55 | 2.1E−05 | 1.4E−03 | IFNAR1 | 27 | 5 | 0.77 | 2.7E−11 | 1.4E−09 |
| TNFSF8 | 28 | 410 | 0.55 | 1.2E−05 | 1.1E−03 | F2RL2 | 28 | 5 | 0.77 | 1.9E−11 | 1.1E−09 |
| CCDC36 | 29 | 576 | 0.55 | 1.5E−05 | 1.2E−03 | CXCR6 | 29 | 1 | 0.77 | 3.1E−11 | 1.5E−09 |
| MRC1 | 30 | 195 | 0.55 | 1.3E−05 | 1.1E−03 | KLHL6 | 30 | 6 | 0.77 | 3.3E−12 | 2.6E−10 |
| CD27 | 31 | 550 | 0.54 | 3.0E−05 | 1.5E−03 | SERPINA5 | 31 | 12 | 0.77 | 2.0E−11 | 1.1E−09 |
| ADCK4 | 32 | 28 | 0.54 | 2.1E−05 | 1.4E−03 | PLA2R1 | 32 | 31 | 0.77 | 6.6E−12 | 4.6E−10 |
| SOWAHA | 33 | 154 | 0.54 | 2.2E−05 | 1.4E−03 | MYCBPAP | 33 | 3 | 0.76 | 4.5E−12 | 3.3E−10 |
| F2RL2 | 34 | 436 | 0.54 | 3.7E−05 | 1.7E−03 | BPIFB2 | 34 | 5 | 0.76 | 7.6E−12 | 4.8E−10 |
| WDR66 | 35 | 302 | 0.54 | 2.1E−05 | 1.4E−03 | TLR7 | 35 | 114 | 0.76 | 1.4E−11 | 8.3E−10 |
| TRADD | 36 | 596 | 0.54 | 2.6E−05 | 1.5E−03 | CCDC190 | 36 | 19 | 0.76 | 2.4E−10 | 6.2E−09 |
| RELA | 37 | 70 | 0.53 | 2.8E−05 | 1.5E−03 | KMT2D | 37 | 95 | 0.76 | 7.1E−12 | 4.8E−10 |
| SLC10A6 | 38 | 533 | 0.53 | 3.0E−05 | 1.5E−03 | FSCB | 38 | 130 | 0.76 | 6.5E−11 | 2.6E−09 |
| IL23A | 39 | 383 | 0.53 | 4.7E−05 | 1.7E−03 | CD27 | 39 | 19 | 0.76 | 2.7E−11 | 1.4E−09 |
| TNFSF18 | 40 | 656 | 0.53 | 5.8E−05 | 1.8E−03 | SNX11 | 40 | 24 | 0.76 | 7.3E−12 | 4.8E−10 |

**Note:**
The top two percent (2%) of ERCs are shown for ACE2 and GEN1, ranked by descending ρ value. The table illustrates how reciprocal ranks can differ between proteins with significant evolutionary correlations, depending on how interconnected proteins are. GEN1 has many partners which rank GEN1 highly in their respective ERCs. Also indicated in the table are examples of reciprocal rank correlations in which both partners rank the other in their top 20 (indicated by bold and asterisks). These are used to construct reciprocal rank protein interaction networks.

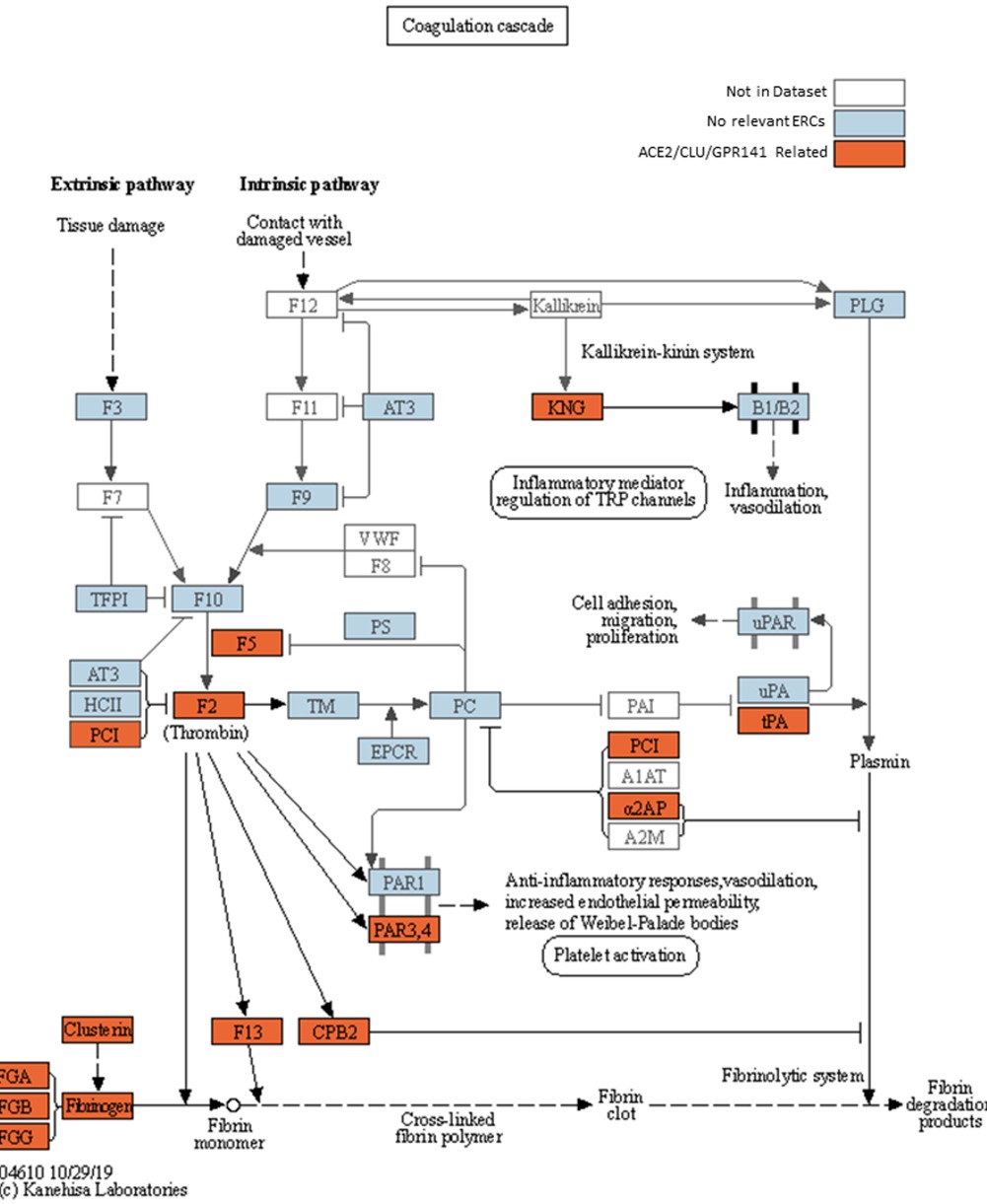

**Figure 2 Modified KEGG coagulation pathway.** KEGG Coagulation cascade pathway (*Kanehisa & Goto, 2000*), with ACE2-CLU-GPR141 associated proteins (based on presence in any of their top 2% ERCs or in the ACE2 CRR network) indicated in orange. The KEGG pathway has been supplemented to indicate the three fibrinogen proteins and clusterin associations previously discussed. Note the alternate protein names: PAR3,4 = F2RL2 & F2RL3 = Thrombin receptors; α2AP = Alpha-2-anti-plasmin = SERPINF2; PLAT = tPA, and PCI = SERPINA5 = Protein C Inhibitor.

whose primary functions are resolution of DNA Holliday junctions and DNA damage checkpoint signaling (*Chan & West, 2015*). This protein shows high ERCs and is ranked highly in the ERC sets for many other proteins, suggesting central connectivity. As described further in "The ACE2 Unidirectional Reciprocal Rank (URR) Network", GEN1
shows unexpected enrichments for immune functions, perhaps related to its role in DNA damage checkpoint signaling.

Because our focus is on identifying strong candidate interactions involving ACE2 and its predicted partners, we utilize the rank information to identify proteins with high reciprocal ranks. Specifically, we focus on the strongest reciprocal ranks (RR) defined by ranks of less than or equal to 20 (RR20), which is the highest one percent of each protein's ERCs, and use these to develop reciprocal rank networks ("ERC Reciprocal Rank Networks Implicate Coagulation Pathways and Immunity"). Although speculative, we posit that protein pairs with high reciprocal ranks are likely to be strongly coevolving (*i.e.* both partners evolving reciprocally due to selective pressures acting on interacting domains between them). In contrast, protein pairs with a significant evolutionary rate correlation only one ranks highly (*e.g.* within the top two percent) in the ERC set of the other, are more likely to be due to "unidirectional" evolution. The rationale is that proteins with many significant ERC partners are under selective pressures primarily from their top evolutionary partners, whereas other interactors evolve primarily in response to the forces shaped by their stronger partner(s). We emphasize that this interpretation is speculative, and requires further exploration to determine what factors shape reciprocal ERC ranks between proteins.

The view that ERCs are detecting protein interactions relevant to COVID-19 is further supported by the analysis of ACE2 reciprocal rank ERC networks ("ERC Reciprocal Rank Networks Implicate Coagulation Pathways and Immunity"). Noteworthy in this regard are additional proteins in the coagulation pathway, such as Coagulation Factor V (F5), Fibrinogen Alpha Chain (FGA), Fibrinogen Beta Chain (FGB), and Fibrinogen Gamma Chain (FGG). Thrombosis (blood clotting) is a major pathology of COVID-19 (*Gupta et al., 2020*). Connections of ACE2 with the proteins above could relate to severe blood clotting problems in COVID-19 infections. ACE2 networks also show strong enrichments of cytokine signaling, viral (and pathogen) infections, and inflammatory response terms (File 3), which are clearly relevant to COVID-19 pathologies such as cytokine storms and systemic inflammation.

In yet other cases, we have found proteins with significant ACE2 ERCs or ACE2 network connections, but for which there is little functional information, such as GPR141. We can use their ERCs to suggest possible functions for future investigation. Finally, ERCs for proteins of known function (such as F5 and GEN1) indicate likely additional roles, suggesting these proteins have unrecognized "moonlighting" functions (*Jeffery, 1999*).

Below, we first describe proteins of interest to which ACE2 has significant ERCs, summarize aspects of their known biological functions, and examine significantly enriched functional categories for these ERCs. We then build and evaluate two different networks for ACE2 interacting proteins ("ERC Reciprocal Rank Networks Implicate Coagulation Pathways and Immunity"), one of which reveals connections to coagulation pathways and the other to cytokine-mediated signaling, viral response, and immunity. Finally, we discuss the potential implications of these predicted ACE2 interactions to COVID-19 pathologies and propose some specific hypotheses that emerge from this analysis.

## Top ERC interactions link ACE2 to COVID pathologies

To investigate protein associations of ACE2, we first determined the protein enrichment categories for its top 2% ERC proteins (based on Spearman rank correlation coefficients, ρ) using the gene set enrichment package Enrichr (*Xie et al., 2021*) (Table 2). The top two KEGG_2019_Human enrichments are for complement and coagulation cascade related (FDR = 2.0E−03) and cytokine-cytokine receptor interaction related (FDR = 2.0E−03) terms. This finding is consistent with two hallmarks of COVID-19 pathology, abnormal systemic blood-clotting (thrombosis) and cytokine storms (*Coperchini et al., 2020*; *Fei et al., 2020*). Additionally, several terms related to viral/bacterial-specific infection are significantly enriched, such as Tuberculosis (FDR = 1.4E−02), HPV infection (FDR = 1.4E−02), measles (FDR = 2.4E−02) and Hepatitis C (FDR = 3.1E−02). Gene Ontology Biological Process also shows enrichment for tumor necrosis factor (TNF) pathways, including the signaling pathway (FDR = 3.9E−03) and cellular responses (FDR = 1.6E−02). Additional terms are shown in Table 2.

The ACE2 ERC analysis indicates that ACE2 is "coevolving" with proteins involved in the complement and coagulation pathways, cytokine signaling, TNF, and pathogen response pathways. Here, we summarize results and background information on some of the key proteins among ACE2's ERCs (more extended summaries of each protein are in the Supplemental Text).

Among ACE2's strongest ERCs are proteins involved in immunity. For example, XCR1 (X-C Motif Chemokine Receptor 1) is ACE2's 2nd top-ranked ERC (ρ = 0.67, FDR = 6.2E−05). It is a chemokine XCL1 receptor involved in immune response to infection and inflammation (*Lei & Takahama, 2012*). Strikingly, the *Severe Covid-19 GWAS Group (2020)* detected a small genomic region containing six genes that significantly associate with severe COVID-19, one of which is XCR1. Our finding that XCR1 is ACE2's 2nd highest ERC interactor lends independent support for a relationship between COVID-19 and XCR1. Furthermore, it suggests that an interaction between ACE2 and XCR1 could be involved in COVID-19 pathologies. To our knowledge, there are no other reports of interactions between these two proteins.

Another striking connection of ACE2 ERC to immunity is through IFNAR2 (Interferon alpha/beta receptor 2), which has a highly significant ACE2 ERC correlation (ρ = 0.62, FDR = 6.1E−04). IFNAR2 forms part of an important receptor complex with IFNAR1 (*Thomas et al., 2011*) involved in interferon signaling through the JAK/STAT pathway to modulate immune responses. IFNAR2 has been implicated in severe COVID-19, based on mendelian randomization, genome-wide associations, and gene expression changes (*Liu et al., 2021*; *Pairo-Castineira et al., 2021*). Our data provide independent support for a role, possibly mediated through ACE2 interactions. Interferon pathways are important in antiviral defense, but also can contribute to cytokine storms and COVID-19 pathologies (*McKechnie & Blish, 2020*). Other immune-related proteins with high ERC connections to ACE2 include TLR8 (Toll-like Receptor 8), FAM3D (FAM3 metabolism regulating signaling molecule D), and PLA2G7 (phospholipase A2 group VII).

**Table 2 Enrichment categories for ACE2's top 2% proteins by ERC.**

| Enrichr gene set | Term | FDR P-value | Odds ratio | Gene list |
|---|---|---|---|---|
| KEGG_2019_Human | Complement and coagulation cascades | 2.03E−03 | 25.9 | CLU, F2RL2, F5, SERPINA5 |
| KEGG_2019_Human | Cytokine-cytokine receptor interaction | 2.03E−03 | 10.5 | TNFSF18, IFNAR2, XCR1, IL23A, CD27, TNFSF8 |
| KEGG_2019_Human | Tuberculosis | 1.38E−02 | 11.0 | IL23A, TRADD, MRC1, RELA |
| KEGG_2019_Human | Human papillomavirus infection | 1.38E−02 | 7.6 | IFNAR2, COL4A4, TRADD, COL4A5, RELA |
| KEGG_2019_Human | Protein digestion and absorption | 1.38E−02 | 16.3 | ACE2, COL4A4, COL4A5 |
| KEGG_2019_Human | Pathways in cancer | 1.38E−02 | 5.7 | IFNAR2, AR, IL23A, COL4A4, COL4A5, RELA |
| KEGG_2019_Human | Small cell lung cancer | 1.38E−02 | 15.8 | COL4A4, COL4A5, RELA |
| KEGG_2019_Human | Amoebiasis | 1.38E−02 | 15.3 | COL4A4, COL4A5, RELA |
| KEGG_2019_Human | AGE-RAGE signaling pathway in diabetic complications | 1.38E−02 | 14.6 | COL4A4, COL4A5, RELA |
| KEGG_2019_Human | Toll-like receptor signaling pathway | 1.39E−02 | 14.0 | IFNAR2, TLR8, RELA |
| KEGG_2019_Human | Sphingolipid signaling pathway | 1.85E−02 | 12.2 | CERS4, TRADD, RELA |
| KEGG_2019_Human | Relaxin signaling pathway | 2.18E−02 | 11.2 | COL4A4, COL4A5, RELA |
| KEGG_2019_Human | Measles | 2.38E−02 | 10.5 | IFNAR2, TRADD, RELA |
| KEGG_2019_Human | Hepatitis C | 3.05E−02 | 9.3 | IFNAR2, TRADD, RELA |
| KEGG_2019_Human | Cocaine addiction | 3.05E−02 | 19.7 | PDYN, RELA |
| KEGG_2019_Human | PI3K-Akt signaling pathway | 4.17E−02 | 5.5 | IFNAR2, COL4A4, COL4A5, RELA |
| KEGG_2019_Human | Kaposi sarcoma-associated herpesvirus infection | 4.17E−02 | 7.7 | IFNAR2, TRADD, RELA |
| KEGG_2019_Human | Inflammatory bowel disease (IBD) | 4.34E−02 | 14.7 | IL23A, RELA |
| KEGG_2019_Human | Epstein-Barr virus infection | 4.34E−02 | 7.1 | IFNAR2, TRADD, RELA |
| KEGG_2019_Human | Adipocytokine signaling pathway | 4.34E−02 | 13.8 | TRADD, RELA |
| KEGG_2019_Human | RIG-I-like receptor signaling pathway | 4.34E−02 | 13.6 | TRADD, RELA |
| KEGG_2019_Human | Pertussis | 4.85E−02 | 12.5 | IL23A, RELA |
| GO_Biological_Process_2018 | tumor necrosis factor-mediated signaling pathway (GO:0033209) | 3.86E−03 | 21.0 | TNFSF18, TRADD, CD27, TNFSF8, RELA |
| GO_Biological_Process_2018 | cellular response to tumor necrosis factor (GO:0071356) | 1.63E−02 | 13.1 | TNFSF18, TRADD, CD27, TNFSF8, RELA |
| GO_Biological_Process_2018 | immunoglobulin mediated immune response (GO:0016064) | 1.63E−02 | 154.6 | CD27, TLR8 |
| GO_Biological_Process_2018 | B cell mediated immunity (GO:0019724) | 1.63E−02 | 154.6 | CD27, TLR8 |
| GO_Biological_Process_2018 | positive regulation of NF-kappaB transcription factor activity (GO:0051092) | 1.85E−02 | 15.6 | TNFSF18, TRADD, CLU, RELA |
| GO_Biological_Process_2018 | I-kappaB kinase/NF-kappaB signaling (GO:0007249) | 2.26E−02 | 26.3 | TRADD, TLR8, RELA |
| GO_Biological_Process_2018 | cytokine-mediated signaling pathway (GO:0019221) | 3.11E−02 | 5.7 | TNFSF18, IFNAR2, IL23A, TRADD, CD27, TNFSF8, RELA |

(Continued)

| Table 2 (continued) | | | | |
|---|---|---|---|---|
| Enrichr gene set | Term | FDR P-value | Odds ratio | Gene list |
| GO_Biological_Process_2018 | regulation of inflammatory response (GO:0050727) | 3.11E−02 | 11.9 | ACE2, IL23A, PLA2G7, RELA |
| GO_Biological_Process_2018 | positive regulation of defense response (GO:0031349) | 3.26E−02 | 20.0 | IL23A, TLR8, PLA2G7 |
| WikiPathways_2019_Human | Complement and Coagulation Cascades WP558 | 2.38E−02 | 25.8 | CLU, F5, SERPINA5 |
| WikiPathways_2019_Human | EBV LMP1 signaling WP262 | 4.24E−02 | 44.2 | TRADD, RELA |
| WikiPathways_2019_Human | Toll-like Receptor Signaling Pathway WP75 | 4.24E−02 | 14.2 | IFNAR2, TLR8, RELA |
| WikiPathways_2019_Human | Toll-like Receptor Signaling WP3858 | 4.36E−02 | 32.0 | TLR8, RELA |
| WikiPathways_2019_Human | miRNAs involvement in the immune response in sepsis WP4329 | 4.95E−02 | 26.5 | TLR8, RELA |
| WikiPathways_2019_Human | Regulation of toll-like receptor signaling pathway WP1449 | 4.96E−02 | 10.4 | IFNAR2, TLR8, RELA |

**Note:**
Key enrichments include complement and coagulation cascades, cytokine-cytokine signaling, and different pathogen infections.

Coagulation pathway proteins figure prominently in ACE2 ERC-predicted protein interactions (Table 3, Fig. 2). This is reflected both in significant enrichment for coagulation cascade proteins in the top 2% strongest ACE2 ERCs (Table 2) and the strong reciprocal rank network for ACE2 ("ERC Reciprocal Rank Networks Implicate Coagulation Pathways and Immunity", Fig. 3). The finding has obvious potential implications to a hallmark pathology of COVID-19, systemic coagulopathy (*Wright et al., 2020*; *Medcalf, Keragala & Myles, 2020*). A list of coagulation and blood-related proteins associated with ACE2 is presented in Table 3. Among ACE2's top 2% ERCs associated with coagulation pathway are Coagulation Factor V (F5), Protein C inhibitor (SERPINA5 aka PCI), and Thrombin Receptor 2 (F2RL2) (Table 1).

Also relevant to coagulopathy are Clusterin (CLU) and the orphan G protein-coupled receptor 141 (GPR141). The chaperone protein CLU has a soluble form that circulates in the blood and is part of the "cleaning squad" that clears misfolded extracellular proteins for delivery to lysosomes and degradation (*Itakura et al., 2020*; *Sánchez-Martín & Komatsu, 2020*). It is the 3rd highest ACE2 ERC ($\rho = 0.63$, FDR = 1.5E−04), and these two proteins show strong reciprocal ranks (3, 8), likely supporting biological interactions. Relevant to this point is that both ACE2 and CLU have soluble forms that circulate in the blood (*Itakura et al., 2020*). Of direct relevance to COVID-19 and possible ACE2-CLU protein interactions, *Singh et al. (2021)* found in cells infected with different coronaviruses (SARS-CoV-2, SARS-CoV, and MERS-CoV), only two genes were found to be differentially expressed in all three, with CLU being one.

CLU's top 2% strongest ERCs show highly significant enrichment for terms relating to coagulation cascades and clot formation (File 3, *e.g.* "Complement and coagulation cascades", FDR = 6.3E−12), as well as significant terms that are relevant to immunity, such as "Immune system" (FDR = 4.8E−03) and "activated immune cell type" (FDR = 3.4E−05).

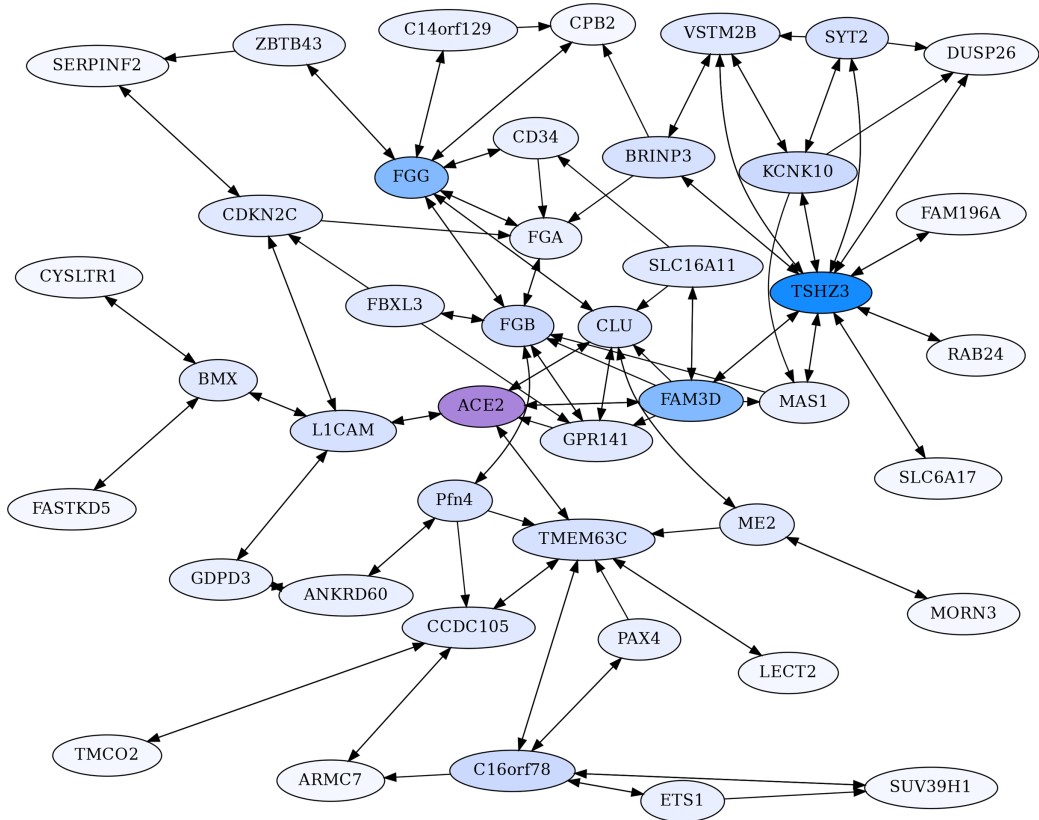

**Figure 3 ACE2 centric reciprocal rank (CRR) network.** Proteins with ERC reciprocal ranks ≤20 are shown by double-headed arrows, and unidirectional ranks ≤20 connecting to the RR backbone are indicated by single-headed arrows. ACE2 has extensive connections to coagulation proteins mediated primarily through Clusterin (CLU) and GPR141. ACE2 is highlighted in purple, and blue shading intensity indicates the level of reciprocal connectivity for different proteins.

Among its top ERC proteins relevant to coagulation process are Coagulation Factor V (F5, ρ = 0.67, FDR = 9.1E−06, rank 3), Fibrinogen Gamma chain (FGG, ρ = 0.59, FDR = 1.7E−04, rank 18), Coagulation Factor XIII B chain (F13B, ρ = 0.63, FDR = 2.8E−05, rank 19), and Fibrinogen Alpha chain (FGA, ρ = 0.57, FDR = 2.9E−04, rank 27) (Fig. 3, File 1). Notably, fibrinogen is a major binding "client" of Clusterin in stressed plasma (*Wyatt & Wilson, 2010*). Little is known about GPR141; however, the ERC analysis suggests an important role in blood coagulation. Among GPR141's top ERC proteins relevant to coagulation process are Kininogen 1 (KNG1, ρ = 0.60, FDR = 9.3E−04, rank 5), Plasminogen Activator (PLAT, ρ = 0.58, FDR = 6.5E−04, rank 6), Thrombin (Coagulation Factor II or F2, ρ = 0.58, FDR = 6.5E−04, rank 7), Fibrinogen Beta chain (FGB, ρ = 0.57, FDR = 6.5E−04, RR 11, 11), Complement C1s (C1S, ρ = 0.54, FDR = 1.6E−03, rank 22), F2R-like thrombin (also called trypsin receptor 3; F2RL3, ρ = 0.52, FDR = 2.6E−03, rank 37), and Coagulation Factor V (F5, ρ = 0.52, FDR = 1.7E−03, rank 39) (Fig. 2, File 1).

GPR141 has a highly significant ERC to CLU, with these two proteins being each other's first ranking ERCs (ρ = 0.68, FDR = 9.1E−06, RR 1,1). The pattern suggests a strong biological interaction, although none is described in the literature. The result supports

**Table 3 ACE2-derived coagulation and blood-related proteins.**

| Name | Full name | Brief description |
| --- | --- | --- |
| ACE2 | Angiotensin-Converting Enzyme 2 | Catalyzes the cleavage of angiotensin I to angiotensin 1-9 and angiotensin II to angiotensin 1-7 (*Burrell et al., 2004*) |
| FGA | Fibrinogen alpha chain | Bind to FGB and FGG to form fibrinogen, used to form blood clots (*Mosesson, 2005*) |
| FGB | Fibrinogen beta chain | Bind to FGA and FGG to form fibrinogen, used to form blood clots (*Mosesson, 2005*) |
| FGG | Fibrinogen gamma chain | Bind to FGA and FGB to form fibrinogen, used to form blood clots (*Mosesson, 2005*) |
| CPB2 | Carboxypeptidase B2 | Inhibits fibrinolysis (*Leenaerts et al., 2018*) |
| SERPINF2 | Serpin family F member 2 (alpha-2-antiplasmin) | Inhibits Plasmin, a protein involved in fibrinolysis (*Kanehisa & Goto, 2000*) |
| CD34 | CD34 molecule | Associated with hematopoiesis and stem cells (*Fina et al., 1990*) |
| CLU | Clusterin | Binds to Fibrinogen (*Wyatt & Wilson, 2010*) |
| MAS1 | MAS1 Proto-Oncogene, G Protein-Coupled Receptor | Receptor for angiotensin-(1-7) (*Burrell et al., 2004*) |
| FAM3D | FAM3 Metabolism Regulating Signaling Molecule D | Implicated in inflammatory responses in the gastrointestinal tract and is a chemoattractant for neutrophiles and monocytes (*Peng et al., 2016*) |
| GPR141 | G Protein-Coupled Receptor 141 | High expression in blood, granulocytes, Kupfer cells, and macrophages (*Stelzer et al., 2016*) |
| TMEM63C | Transmembrane Protein 63C | Interacts with angiotensin II (*Eisenreich et al., 2020*) |
| LECT2 | Leukocyte Cell-derived Chemotaxin 2 | Involved in macrophage activation, insulin resistance and diabetes, and neutrophil chemotaxis (*Yamagoe et al., 1996*; *Zhang et al., 2018*; *Takata et al., 2021*) |
| ETS1 | ETS proto-oncogene 1, transcription factor | Transcription factor involved in cytokine/chemokine processes and angiogenesis (*Stelzer et al., 2016*) |
| ZBTB43 | Zinc Finger and BTB Domain containing 43 | Associated with Diamond-Blackfan Anemia 4, in which the bone marrow is unable to make enough red blood cells to carry oxygen (*Stelzer et al., 2016*) |
| COL4A4 | Collagen Type IV Alpha 4 | Subunit of Collagen Type 4, which are a part of the basement membrane which resides between epithelial cells (*Stelzer et al., 2016*) |
| F13B | Coagulation Factor XIII B chain | Stabilizes F13A subunits, while it does not have enzymatic abilities it is thought to be a plasma carrier molecule (*Stelzer et al., 2016*) |
| AMOT | Angiomotin | Associated with angiogenesis and endothelial cell movement (*Bratt et al., 2005*; *Aase et al., 2007*) |
| PDYN | Prodynorphin | Inhibits vasopressin secretion (*Yamada et al., 1988*) |

**Note:**
Coagulation and blood-related proteins in the ACE2 CRR and URR Networks as well as the top 1% ACE2 ERC list.

investigating functional interactions between CLU and GPR141, based upon their high ERC and reciprocal ranks. Our network analysis below ("ERC Reciprocal Rank Networks Implicate Coagulation Pathways and Immunity") further supports extensive interconnections among ACE2, Clusterin, GPR141, and coagulation pathway proteins, implicating the protein interaction pathway as a possibly significant contributor to disruption of coagulation in COVID-19 disease. Coagulation cascade proteins found in the ACE2's top 2% ERCs, ACE2 reciprocal rank network, and Clusterin-GPR141 associated proteins are highlighted in Fig. 2.

Androgen Receptor (AR, $\rho = 0.57$, FDR = 8.8E−04, rank 13) is the receptor for the male hormone androgen. It plays a major role in reproductive system development, somatic differentiation, and behavior (*Matsumoto et al., 2008*). Androgen-AR signaling induces ACE2 (*Wu et al., 2020*), while knockdowns of AR result in downregulation of ACE2 (*Samuel et al., 2020*). AR agonists also reduce SARS-CoV-2 spike protein-mediated cellular entry (*Deng et al., 2021*). Additionally, AR is associated with COVID-19 comorbidities (*Dolan et al., 2020*), and recently implicated in the severity of COVID-19 in women with polycystic ovarian syndrome, a disorder associated with high androgen levels and androgen sensitivity (*Gotluru et al., 2021*). Our ERC finding indicating ACE2 and AR coevolution suggests regulatory feedback between these two proteins, which could be relevant to COVID-19 severity and other sex differential pathologies, such as cardiovascular disease (*Viveiros et al., 2021*).

Other notable significant ACE2 ERCs (Table 1) include Metabolism regulating signaling molecule D (FAM3D), Transmembrane-protein 63C (TMEM63C); Collagen Type IV Alpha 4 (COL4A4), L1 cell adhesion molecule (L1CAM), and ITPRIP-like 2 (ITPRIPL2). More detailed information on these and other proteins mentioned in this section is provided in "ERC Reciprocal Rank Networks Implicate Coagulation Pathways and Immunity" and the Supplemental Text.

## ERC reciprocal rank networks implicate coagulation pathways and immunity

As mentioned previously, two proteins with a significant evolutionary rate correlation (ERC) may often "rank" each other differently in their respective top ERC connections. This occurs because some proteins have more extensive ERC connections than others. High reciprocal ERC ranks between protein pairs may be more indicative that they are under strong coevolutionary pressure in their sequence and function. We have thus found it useful to evaluate these reciprocal rank connections as a network. The rationale is that such proteins are likely to be reciprocally evolving ("coevolving"). To build reciprocal rank networks, we use protein pairs that reciprocally share ranks less than or equal to 20 (RR20), which are the top one percent for each protein's respective ERC set.

A core ACE2 reciprocal rank network was generated by building reciprocal rank connections (RR20) outward of ACE2, to provide a backbone set of RR20 protein connections. The backbone was expanded on by adding the RR20 connections of the non-ACE2 backbone proteins. Unidirectional ERCs (≤ rank 20) were then added between proteins within the RR set to produce an ACE2 Core Reciprocal Rank (CRR) Network (Fig. 3). The network is designed to capture features of ACE2's protein interactions as revealed by the strong reciprocal evolutionary correlations among proteins.

ACE2 also has highly significant ERCs to proteins that do not rank ACE2 within their top 1% of ERCs, due to those proteins having more protein interactions with higher ERCs. A second network was therefore generated using ACE2's top ten unidirectional ERCs, followed by calculating the RR20 associations for those proteins. This second network is referred to as the ACE2 Unidirectional Reciprocal Rank (URR) Network (Fig. 4).
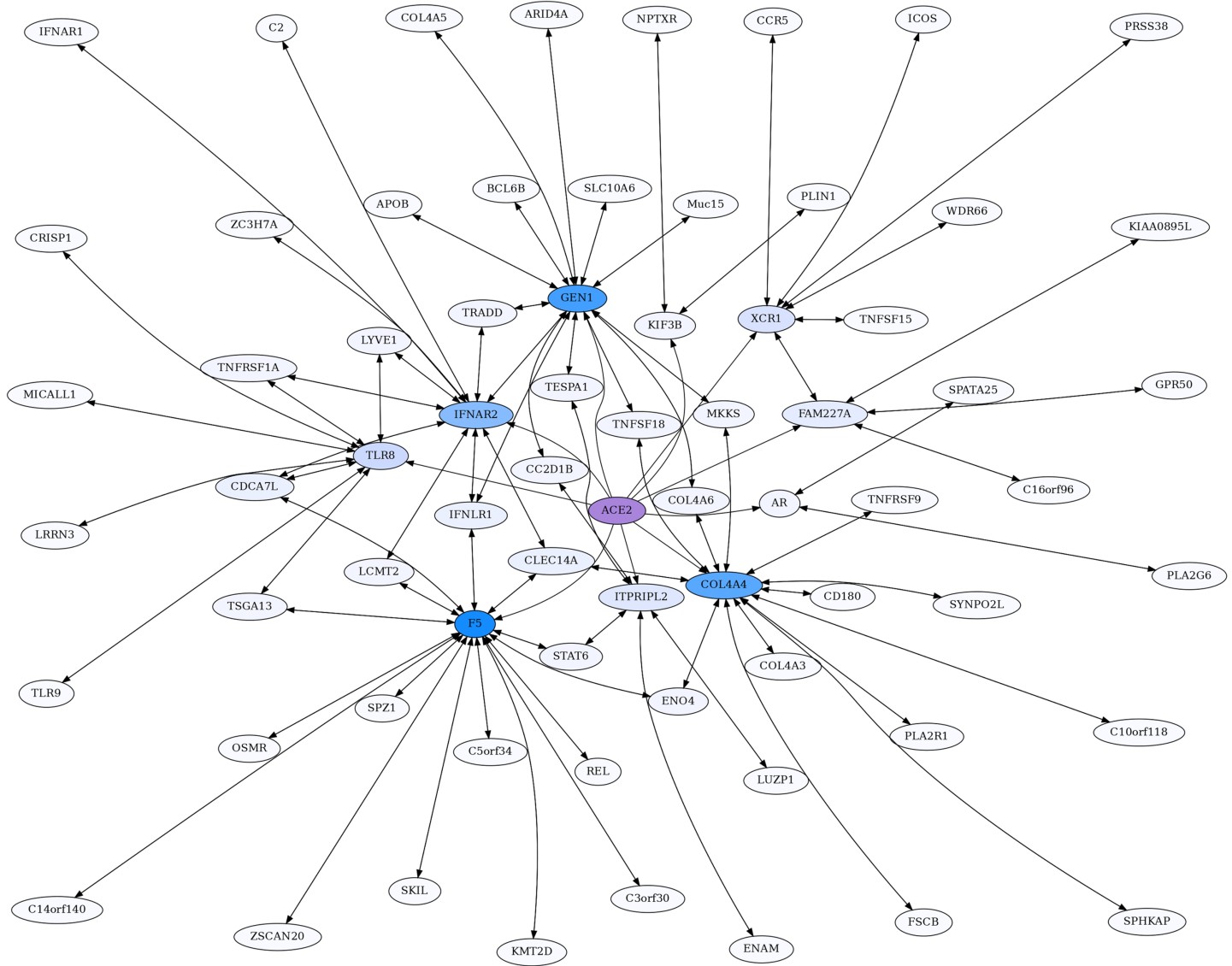

**Figure 4 ACE2 unidirectional reciprocal rank (URR) network.** ACE2's top 10 unidirectional ERC proteins for a web of reciprocal rank (RR20) connections. The network is particularly enriched for cytokine signaling and immunity. Highly interconnected proteins include COL4A5, F5, GEN1, and IFNAR2. ACE2 is highlighted in purple, and blue shading intensity indicates the level of reciprocal connectivity for different proteins.

These are presented below. In general, the reciprocal ranks analysis lends credence to our proposition that ERCs reveal real biological interactions, as well as providing predictions for novel protein interactions possibly of importance to COVID-19 pathologies and protein-interaction networks.

### The ACE2 core reciprocal rank (CRR) network

The CRR network (Fig. 3) is designed to capture essential features of ACE2's protein interactions as revealed by the strong reciprocal correlations among proteins.

The most striking aspects of the ACE2 CRR Network are extensive connections to the coagulation pathway and blood-associated proteins (Fig. 3, Table 3). This could be relevant to COVID-19 due to extensive clotting pathologies and stroke associated with COVID-19 (Bonaventura et al., 2021), as well as microvascular clotting and the apparent shut-down of fibrinolysis (Wright et al., 2020). Extensive blood coagulation of COVID-19 patients can even lead to clogging of dialysis equipment (Rabb, 2020). This hallmark pathology of COVID-19 indicates a disruption in coagulation and fibrinolysis pathways, and our findings of extensive network connections between ACE2 and coagulation-fibrinolysis pathway proteins could be relevant. The predicted novel protein interactions detected here may also have implications more generally to circulatory system homeostasis, including regulation of blood pressure and coagulation.

ACE2 connects to coagulation pathway proteins through F5, CLU, FAM3D, and GPR141 (Figs. 2 and Fig. 3). CLU-GPR141 form a high RR ERC (ranks 1,1), strongly suggesting coevolution of these proteins and physical/functional interactions. Both CLU and GPR141 then connect to the fibrinogen proteins FGB and FGG. FGA, FGB, and FGG are the three protein components that make up fibrinogen, which during the clotting process are converted into fibrin monomers, which subsequently cross-link to form the fibrin clot (Mosesson, 2005). All three proteins form an RR20 triad, indicating protein coevolution. FGG is a hub for RR ERCs to several other proteins (e.g. CD34, CPB2, C14or129, and ZBTB43). ZBTB43 is noteworthy, as it is associated with the blood diseases Diamond-Blackfan Anemia 4 and Hemochromatosis Type 2 (Stelzer et al., 2016). The former disrupts red blood cell formation in the bone marrow and the latter causes iron accumulation in the body. In terms of tissue distribution, ZBTB43 is enhanced in bone marrow (Uhlén et al., 2015). Cellularly, it is found mainly in nucleoplasm and nucleoli, suggesting regulatory functions, as might be expected for a transcription factor-like zinc finger domain protein. Most noteworthy, Mamoor (2020) has shown that ZBTB43 is differentially expressed in human microvascular endothelial cells and human cell cultures infected with coronaviruses (e.g MERS-CoV and human coronavirus 229E). So, this is yet another member of the ACE2 protein Network which is implicated in coronavirus infection. In turn, ZBTB43 has a RR connection with SERPINF2, which enhances clotting by inhibiting plasmin, an enzyme that degrades fibrin, the main component of clots. Mutations in SERPINF2 can cause severe bleeding disorders and upregulation of SERPINF2 is implicated in COVID-19 patient thrombosis (Jain et al., 2021; Lazzaroni et al., 2021). In turn, CPB2 (Carboxypeptidase B2) is a thrombin-activated inhibitor of fibrinolysis, and therefore enhances clotting stability (Leenaerts et al., 2018), and also plays a role in activating the complement cascade (Morser et al., 2018; Leung & Morser, 2018).

FAM3D is a cytokine for neutrophils and monocytes in peripheral blood which may interact with ACE2 based on their reciprocal ranking. ACE2 is its 2nd ranking ERC. Although ACE2 does not have a significant ERC to F13B (also known as Coagulation Factor XIII B Chain), it is FAM3D's top-ranking ERC. F13B functions to stabilize clotting through cross-linking of fibrin (Stelzer et al., 2016). Thus, the predicted interaction of FAM3D and F13B may be relevant to the coagulation pathway.
Blood pressure and vasoconstriction regulation also show functional enrichment in the CRR network. Naturally, ACE2 is a crucial component of the Renin-Angiotensin System (RAS), which converts angiotensin II to angiotensin (1–7). This, in turn, binds to the MAS1 receptor, promoting vasodilation and reduced blood pressure. As seen in Figure 2, MAS1 is part of the ACE2 CRR network. Although not significantly correlated with ACE2 directly, it has significant RR connection to TSHZ3 ($\rho$ = 0.52, FDR = 7.8E−03, ranks 11, 4) and is FAM3D's 19th ranking ERC ($\rho$ = 0.49, FDR = 1.5E−02). Biologically MAS1 and ACE2 are key elements promoting vasodilation in the renin-angiotensin system (RAS) (*Burrell et al., 2004*). Thus, the ERC RR network detects biologically significant connections of ACE2 to RAS signaling *via* the MAS1 receptor of angiotensin-(1–7). *Samavati & Uhal (2020)* posit that the loss of ACE2 due to SARS-CoV-2 infection reduces MAS1 signaling and increases AT1 & AT2 signaling *via* higher levels of angiotensin 2, promoting vasoconstriction, fibrosis, coagulation, vascular and cardio injury, and ROS production. Similar arguments are made by *Sriram & Insel (2020)*. ACE2 and MAS1 do not have a signature of protein coevolution, even though they interact indirectly biologically through the short seven amino acid signaling peptide Ang (1–7). In contrast, MAS1 has a significant RR with TSHZ3 (mentioned above). A biological connection between these proteins is not obvious, although the high ERC reciprocal ranks suggest possible interactions worth further investigation. Additionally, TMEM63C is one of four proteins that form a reciprocal rank ERC association with ACE2 (Fig. 2). It functions in osmolarity regulation and like ACE2, interacts with angiotensin II, possibly reducing damage to kidney podocytes (*Eisenreich et al., 2020*).

FBXL3 has a RR20 connection to FGB and ranks GPR141 in its top 2%. This protein is a component of circadian rhythm regulation (*Busino et al., 2007*). Many aspects of the cardiovascular system have circadian cycling such as heart rate, blood pressure, and fibrinolysis (*Reilly, Westgate & FitzGerald, 2007*). Endogenous oscillators in the heart, endothelial cells, and smooth muscles may play significant roles in these cycles (*Reilly, Westgate & FitzGerald, 2007*), and the CRR network suggests that interactions between FBXL3 and FGB could play a role in circadian aspects of fibrinolysis.

CD34 (Hematopoietic Progenitor Cell Antigen CD34) is believed to be an adhesion protein for hematopoietic stem cells in bone marrow and for endothelial cells (*Fina et al., 1990*). Our ERC analysis indicates connections to coagulation pathway proteins and lipoproteins. In addition to its RR association with FGG ($\rho$ = 0.60, FDR = 2.2E−04, ranks 18,9), CD34 also forms significant reciprocal rank correlations with coagulation factor F2 ($\rho$ = 0.69, FDR = 7.9E−06, ranks 1,6), lipoprotein APOE ($\rho$ = 0.64, FDR = 6.0E−05), lipid droplet-associated protein PLIN1 ($\rho$ = 0.64, FDR = 1.1E−04, ranks 8,7), and inflammation associated pentraxin protein PTX3 ($\rho$ = 0.65, FDR = 6.8E−05, ranks 3,11) (File 1). As expected from these protein associations, CD34's top enriched term is to complement and coagulation cascade (FDR = 1.4E−08). There is also enrichment for HUVEC cells (FDR = 3.1E−05) and Blood Plasma (FDR = 1.7E−04) (File 3).

Additional proteins of interest are discussed further in the Supplemental Materials, including TSHZ3 (a key regulator of airflow and respiratory rhythm control) and L1CAM

(involved in nervous system development and vascular endothelial cell differentiation from stem cells).

Consistent with the descriptions above, the CRR network shows enrichment (full enrichment table in File 3) for negative regulation of blood coagulation (FDR = 4.3E−08), platelet alpha granule-related terms (FDR = 1.7E−05), plasma cell (FDR = 8.3E−4) and blood clot (FDR = 4.5E−02). These enrichments indicate that the network involves protein interactions related to blood clotting pathways. There are also several significantly enriched terms which are driven in part by ACE2, such as regulation of systemic arterial blood pressure by renin-angiotensin (FDR = 1.6E−03), metabolism of angiotensinogen to angiotensin (FDR = 6.9E−03), regulation of blood vessel diameter (FDR = 1.5E−02), and renin-angiotensin system (FDR = 1.8E−02).

### The ACE2 unidirectional reciprocal rank (URR) network

ACE2 also has highly significant ERCs with interacting proteins that are unidirectional, meaning that ACE2 ranks these proteins in its top 2%, but the partner protein does not rank ACE2 within its top 2% due to higher ERC correlations with other partners (Table 1). Some of ACE2's highest-ranking proteins fall into this category, including GEN1 (rank 1), XCR1 (2), IFNAR2 (5) KIF3B (6), and ITPRIPL2 (7), FAM227A (8), TLR8 (9), COL4A4 (10), F5 (12), and AR (13). To focus on strong protein connections in this set, we took the top ten proteins with unidirectional ERCs for ACE2 and then added their reciprocal rank 20 (RR20) partners. The resulting ACE2 Unidirectional Reciprocal Rank (URR) Network contains 69 proteins (Fig. 4).

Notable in the network are many proteins involved in immunity and cytokine signaling, such as IFNAR2 (Interferon alpha/beta receptor 2), XCR1 (X-C Motif Chemokine Receptor 1), and ICOS (Inducible T Cell Costimulator). There are also Toll-Like Receptors TLR8 and TLR9, which stimulate innate immune activity (Forsbach et al., 2011), and Tumor Necrosis Factor related proteins such as TNSFS18, TNTSF15, TNFRSF9, and TNRRSF1A.

Enrichment analysis of the URR network generates 72 significant terms (File 3). The network is highly enriched for cytokine-cytokine receptor interaction (FDR = 6.5E −06), I-kappaB kinase/NF-kappaB signaling (FDR = 1.6E−06), necroptosis (FDR = 3.3E −03), viral infections, such as Human Papillomavirus (FDR = 5.7E−04) and Herpes virus (FDR = 3.5E−03), JAK-STAT and PI3K-AKT signaling pathways, Toll-like receptor signaling, and immune system *Homo sapiens* (FDR = 3.7E−03).

XCR1 is the 2nd highest ACE2 ERC. It is the receptor for chemokine XCL1, which is produced in response to infection and inflammation, and during development of regulatory T cells (*Lei & Takahama, 2012*). Furthermore, XCR1 maps to a region implicated in severe COVID-19 by a genome-wide association study (*Severe Covid-19 GWAS Group, 2020*). As seen in Fig. 4, XCR1 forms a RR subnetwork with six other proteins (ICOS, CCR5, WDR66, TNSFS15, PRSS38 and FAM227A), three of which are known to be involved in immunity. ICOS (Inducible T Cell Costimulator) is reciprocally evolving with XCR1 based on their ERC interaction. It is an inducible T Cell stimulator that is essential for T helper cell responses (*Hutloff et al., 1999*; *Tafuri et al., 2001*).

In addition, ICOS signaling is impaired in COVID-19 patients requiring hospitalization (*Hanson et al., 2020*). The high ERC between ACE2 and XCR1 and high reciprocal ranks of XCR1 to ICOS suggests that the disruption of an ACE2-XCR1 interaction could have a contributory role in COVID-19. C-C Motif Chemokine Receptor 5 (CCR5) forms a significant RR ERC with XCR1 as well. Several studies have implicated CCR5 variation and expression to be associated with COVID-19 severity (*Gómez et al., 2020*; *Hubacek et al., 2021*; *Kasela et al., 2021*), while others have not (*Bernas et al., 2021*). TNFSF15 is a third immune response protein in the XCR1 RR subnetwork that shows elevated expression in patients with severe COVID-19 (*Jain et al., 2021*). We recognize that the involvement of these immune-related proteins in COVID-19 does not require an effect mediated through ACE2. Instead, their protein evolutionary correlations suggest that ACE2 may play a contributory role to COVID-19, possibly through XCR1-related pathways, as suggested by the network analysis.

IFNAR2 is another protein that is highly correlated with ACE2 ($\rho = 0.62$, FDR = 6.1E−04) and is also implicated in severe COVID-19 by GWAS and expression data (*Liu et al., 2021*; *Pairo-Castineira et al., 2021*). It has RR20 ERCs with ten other proteins and is embedded in a complex web of interactions with members of the ACE2 network. Here we draw attention to a few key features. Notably, IFNAR2 and IFNAR1 are RR partners, as expected given that they combine to form the IFN-alpha/beta receptor, which is the receptor for both alpha and beta interferons. IFNAR2 forms a high RR relationship with TNFRSF1A ($\rho = 0.84$, FDR = 4.8E−12, 1,1 reciprocal ranks). This protein is the receptor for TNFα and the pathway affects apoptosis and inflammation regulation. *Jin et al. (2015)* found that ACE2 deletion increases inflammation through TNFRSF1A signaling, lending further support to a functional association between ACE2 and this protein.

GEN1 is the highest-ranking ACE2 ERC protein ($\rho = 0.67$, FDR = 4.2E−05), and it functions as a resolvase of Holliday junctions and a DNA damage checkpoint signaling (*Chan & West, 2015*). Frankly, we are perplexed by the functional significance of ACE2-GEN1 correlated evolution. As observed in the ACE2 network, GEN1 is a highly interconnected protein, with 14 RR20 connections in the network. This result suggests that GEN1 may have additional functions beyond DNA replication. Indeed, although its second-highest RR is to CC2D1B (2,1), a protein involved in mitosis, its highest RR is to Interferon Lambda Receptor 1 (IFNLR1), with an impressive Spearman correlation of $\rho = 0.89$ (FDR = 6.2E−17). As IFNLR1 binds cytokine ligands and stimulates antiviral response, this suggests some feedback mechanism between GEN1 and the immune system, possibly related to its functional role in DNA damage checkpoint signaling. Indeed, its top 2% ERCs show enrichment for multiple viral infection terms (File 3). Therefore, it appears that GEN1 has a "hidden life" that ERC analysis suggests warrants exploration.

The Collagen Type IV A4 subnetwork (Figs. 4, 5) lends further credence to the view that ERCs can detect proteins with likely binding partners. COL4A4 is a component of the Collagen Type IV protein complexes in basement membranes in the extracellular matrix of various tissues, including the kidney glomerulus and vascular endothelial cells, and lung alveoli (*Myllyharju & Kivirikko, 2001*). COL4A4, COL4A3, and COL4A5 complex with

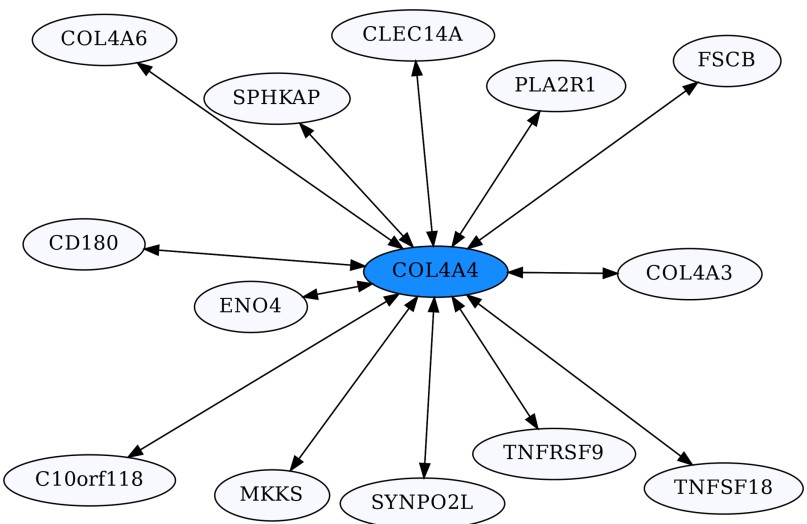

**Figure 5 COL4A4-centric RR20 network.** This network detects reciprocal rank 20 ERCs of different proteins to COL4A4, including other Type IV Collagen proteins known to form complexes with COL4A4.

each other in the basement membranes of kidney glomeruli – mutations in these COL4A proteins are known to cause different kidney disorders (*Torra et al., 2004*; *Wiradjaja, DiTommaso & Smyth, 2010*). Consistent with their expected binding, COL4A4 and COL4A3 are each other's reciprocal best partners (ranks 1,1) and highly correlated with each other ($\rho = 0.88$, FDR = 4.4E−16). Both show highly significant ERCs to COL4A6 (rank 6,5 for COL4A4 $\rho = 0.83$, FDR = 2.1E−12; rank 22,30 for COL4A3 $\rho = 0.78$, FDR = 1.6E−10). Thus, evolutionary rate correlations show highly significant ERCs among Collagen Type IV proteins known to physically interact. A future direction is to use ERCs to more precisely define predicted coevolving protein segments, which could be used to inform docking simulations and experimental studies.

COL4A5 also has significant ERCs to COL4A3 ($\rho = 0.71$, FDR = 2.2E−08) and COL4A4 ($\rho = 0.71$, FDR = 1.7E−08), but these do not qualify as RR20 due to the large number of high ERCs for COL4A5. Interestingly, COL4A5-MUC15 are top-ranking partners (ranks 1,1) with a very high ERC ($\rho = 0.89$, FDR = 3.2E−16). MUC15 is a cell surface protein that is believed to promote cell-extracellular matrix adhesion and it is implicated in affecting influenza infection (*Chen et al., 2019*), which may increase its relevance in the context of COVID-19 infection. ERCs may help to inform candidate domains within each protein that are involved in their expected binding affinity.

Coagulation Factor V (F5) is known for its role in the coagulation cascade. However, F5 is a highly ERC-connected protein, with 43 proteins ranking it in their respective top 5 highest ERCs. This connectedness is also reflected in the RR20 network shown below (Fig. 6). F5 has 16 RR20 connections out of the maximum 20 possible. Although F5 is a vital protein in the coagulation cascade, its top 16 RR connections indicate immune functions, including Interferon λ receptor 1 (IFNLR1; RR 4,10) and Oncostatin M Receptor (OSMR; RR 1,4). This is reflected in the enrichments among its 16 RR proteins

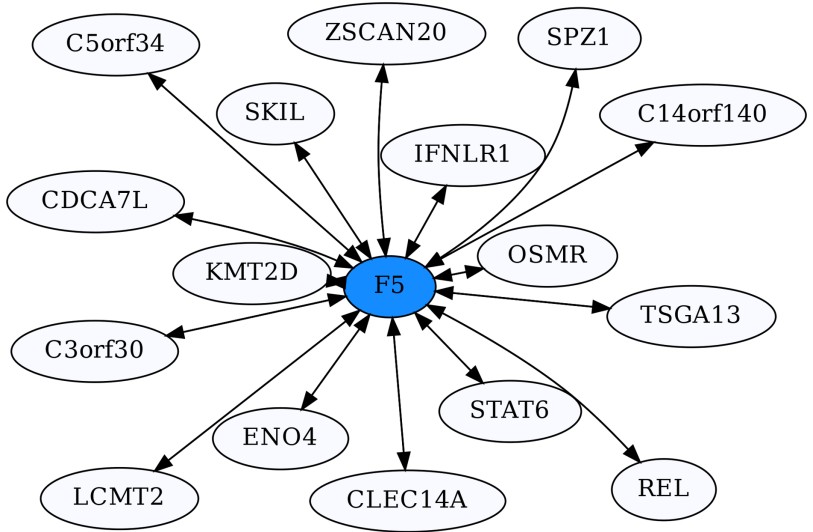

**Figure 6 Coagulation factor V-centric RR20 network.** The network captures strong reciprocal ERCs between F5 and proteins related to immune function such as IFNLR1.

for the JAK-STAT signaling pathway (FDR = 8.7E–03) and response to cytokine (FDR = 2.5E–02). Similarly, the F5 top 2% ERC show enrichments for 54 terms (File 3); notably many related to inflammatory response (FDR = 1.1E–03) and the complement system (FDR = 8.4E–03). The functions of several of F5's RR20 partners are not well known, such as C14orf140 and C5orf34. Their top 2% enrichment suggests cytokine receptor activity (FDR = 2.7E–02) for C14orf140, and Human Complement System (FDR = 1.9E–03) and cytokine receptor activity (FDR = 2.1E–02) for C5orf34.

In conclusion, F5 appears to have a "secret life" of strong protein interactions reflecting moonlighting functions with extensive signaling or modulation roles beyond coagulation regulation.

## ERCs and protein interactions

We postulate that ERCs detect proteins that are coevolving due to functional interactions. Furthermore, we propose that physical binding is an important mechanism contributing to significant ERCs between proteins. This is consistent with anecdotal observations from this study of high reciprocal rank ERCs among the fibrinogen components FGA, FGB, & FGG, the Collagen Type IVA proteins COL4A4, COL4A3, and COL4A6 proteins, and Interferon alpha/beta proteins IFNAR2 and IFNAR1.

To further investigate the role of binding affinity, we examined the mammalian protein complex database CORUM (*Giurgiu et al., 2019*) to determine whether significantly higher Spearman rank correlations (ρ values) are found among proteins within known protein complexes. A set of 139 protein complexes (excluding those with overlapping proteins) were identified which contain at least two members from our ERC data set, for a total of 258 pairwise comparisons. We compared the ρ values of within complex proteins to the median values for proteins outside the complex and found that Spearman rank

correlations of within complex proteins were significantly higher than its between complex values according to Wilcoxon matched signs rank tests (WMRST) under a significance level of $\alpha = 0.05$ ($p = 5.2\mathrm{E}{-}04$), with a median increase of 6.3% (File 11). Many of the complexes contain large numbers of proteins, reducing the probability of direct physical contact between individual members. We therefore also analyzed only proteins from complexes with 5 or fewer members (96 pairs). In this case, the median $\rho$ value increase is 15.8% (WMSRT $p = 6.2\mathrm{E}{-}03$). The results support the view that proteins within known complexes show higher ERCs than between complexes, and further implicate physical contact as a contributor to ERCs. However, other studies have found ERCs between proteins that do not bind to each other, but are involved in shared function, such as metabolic pathways (*Clark, Alani & Aquadro, 2012*). Thus, future research is needed to better understand the different biological drivers of ERCs between proteins.

## DISCUSSION

An overwhelmingly strong pattern is an association between ACE2, its partners, and the proteins involved with coagulation, cytokine signaling, and immunity. For coagulation, this is exemplified by the enrichment for terms related to coagulation pathways in the CRR network, and the presence of the three proteins that form fibrinogen (FGA, FGB, FGG) which constitutes the clotting molecule fibrin. Abnormal clotting and coagulation such as "hypercoagulability" has been observed as a major symptom of COVID-19 infection (*Fei et al., 2020*). Additionally, disseminated intravascular coagulation (DIC) due to COVID-19 has been found more frequently in fatal cases of COVID-19 than non-fatal cases (*Seitz & Schramm, 2020*). *Levi et al. (2020)* have noted that low-grade DIC often seen in COVID-19 is associated with a sudden decrease in plasma fibrinogen before death. This makes the connection with the various fibrinogen subcomponents even more striking. Our network data suggest that ACE2's connection to fibrinogen is mediated through Clusterin and GPR141 (Fig. 3). The chaperone protein Clusterin's role in removing misfolded proteins in the blood and its common association with fibrinogen in blood plasma (*Wyatt & Wilson, 2010*) lend credence to these ERC findings. What remains unclear is the nature of potential functional interactions between ACE2 and Clusterin, but the ERC results suggest that this warrants further attention. The discovery of a strong ERC association of Clusterin and GPR141 is a novel finding, as functional information on GPR141 is largely lacking. ERC analysis indicates that these proteins functionally interact, likely involving coagulation processes.

Another mechanism for ACE2's influence on the coagulation effects of COVID-19, based on ERCs, is through F5. F5 canonically is activated by the same enzyme (Thrombin) that converts fibrinogen into fibrin for clotting (*Omarova et al., 2013*). *Omarova et al. (2013)* further report that inhibition of F5 can enhance an anticoagulant ability of an alternate fibrinogen that utilizes a different isoform of FGG, fibrinogen γ′. Thus, we hypothesize that abnormal coagulation activity may (in part) be driven by disruptions in ACE2-F5 protein interactions, which could reduce anticoagulant feedback mechanisms. F5 is also found to have many significant ERCs outside of the coagulation pathway, connecting to various immunity-related pathways (Fig. 4, File 1). The ERC results for

GPR141 and F5 reveal how ERC analysis may be useful in providing testable hypotheses for functions of understudied proteins, and to investigate additional functional roles on well-studied proteins.

A second major finding is ACE2 protein-protein interactions that connect to cytokine signaling and immunity. "Cytokine storms", an overreaction of the immune system which can lead to inflammation and organ failure, is a second major hallmark of severe COVID-19, and its management is a major target of medical treatment research (*Luo et al., 2020*; *Mangalmurti & Hunter, 2020*). Chemokines are a class of cytokines that act as immune cell attractants (*Coperchini et al., 2020*), and an increase in chemokine production may be characteristic of COVID-19 infection (*Coperchini et al., 2020*). XCR1 is a receptor of XCL1 chemokines, mostly expressed in dendritic cells, and plays a role in cytotoxic immune responses (*Lei & Takahama, 2012*). The XCR1 protein, strikingly, is the second-highest ERC to ACE2 and has already been implicated in severe COVID-19 infection (*Severe Covid-19 GWAS Group, 2020*). While the specific mechanism by which XCR1 might play a role in severe COVID-19 is not yet known, ERC results indicate its role may be mediated by ACE2 with XCR1's ERCs also possibly indicating a broader functional role in coagulation. Excessive Inflammatory response, particularly as a consequence of cytokine storms, is a clear pathology or COVID-19.

Type 1 interferons are among the first types of cytokines produced after viral infection (*García-Sastre & Biron, 2006*; *Sallard et al., 2020*). A component of the type 1 interferon receptor, IFNAR2, is among the strongest ACE2 ERCs, possibly linking ACE2 to the type 1 interferon immunity response. Notably, IFNAR2 has been implicated in severe COVID-19 infection (*Pairo-Castineira et al., 2021*). Since type 1 interferons have shown some initial efficacy in treating COVID-19 infection (*Sallard et al., 2020*), it is possible that the SARS-CoV-2 virus interaction with both receptor and soluble ACE2 interferes with type 1 interferon response, as low levels of type 1 interferons have been found in COVID-19 patients (*Salman et al., 2021*). Another connection of ACE2 with immunity may be mediated by the toll-like receptor TLR8 (a strong ACE2 ERC), among TLRs believed to regulate platelet circulation in response to inflammation (*Beaulieu & Freedman, 2010*) providing possible avenues for interaction with soluble ACE2 in blood. Genetic variants in TLRs (including TLR8) may affect COVID-19 susceptibility (*Lee, Lee & Kong, 2020*). Thus, there are many potential avenues for ACE2 protein interactions contributing to immune dysregulation in COVID-19 disease, which may warrant further investigation given the strong ERC associations of ACE2 with proteins relevant to immunity, although the functional bases of such interactions are unknown. Other ACE2 network ERCs of interest are relevant to kidney disease, cardiovascular disease, male fertility, Alzheimer's disease, and DNA damage checkpoint signaling. These are discussed further in the Supplemental Text.

Overall, the underlying concept behind the evolutionary rate correlation approach (also called evolutionary rate covariance or evolutionary rate coevolution) is that coevolving proteins will show correlated rates of change across evolution and that this reflects functional interactions (*Clark, Alani & Aquadro, 2012*; *Wolfe & Clark, 2015*). Clark and colleagues have developed a web interface (https://csb.pitt.edu/erc_analysis/) to screen for

ERC interactions for Drosophila, yeast, and mammals. Their mammalian data set is based on 33 mammalian species (*Priedigkeit, Wolfe & Clark, 2015*; *Wolfe & Clark, 2015*). We have compared their output for ACE2 to our analyses and found only one overlapping protein (XCR1) between their significant ERCs ($p < 0.05$) and our top 2% ACE2 ERCs. There are many methodological differences between our approaches, including the number and specific mammalian taxa used, the method for calculating protein rates, and the phylogeny used for calculating branch lengths. In addition, their dataset includes 17,487 proteins, whereas our analysis is currently restricted to 1,953 proteins for which we were confident about 1:1 orthology and therefore for which there are minimal paralogy complications. Furthermore, we are uncertain how their database dealt with potential short branch artifacts on ERC calculations. In our case, we found that short branches in the phylogeny resulted in significant correlations between branch time and protein rate, thus both inflating estimated ERCs and introducing branch time as a confounding factor which can lead to spurious correlations, and we removed these by branch trimming.

In another study, *Braun et al. (2020)* applied a "phylogenetic profiling" approach to identify ACE2 interacting proteins relevant to possible drug targets for COVID-19. Phylogenetic profiling generally screens multiple genomes for presence-absence correlations of protein combinations, as a method to detect candidate protein interactions (*Pellegrini et al., 1999*). However, *Braun et al. (2020)* use a modification of the method that also incorporates a BLAST-based distance metric from human ACE2 across taxa ranging from humans to fungi. When we focus on proteins common between our set and their mammalian data set (1,875 proteins), there are three shared proteins among the top 1% for both sets, Androgen Receptor (AR) and Angiomotin (AMOT), and nucleotide excision repair protein homolog MMS19, with no additional proteins in the respective top 2% sets. We suggest that our direct measures of protein evolutionary rates, which utilize aligned sequences and phylogenetic analysis, may be a more sensitive approach for finding evolutionary interactions among proteins in mammals. Obviously, future validation studies are needed to reveal which approaches are most effective at detecting candidate protein interactions, or whether each has its own merits for the detection of different interactions.

Experimental validations of novel ACE2 protein associations predicted by our ERC approach are clearly needed. A necessary first step is to establish whether ACE2 has binding affinities *in vitro* and *in vivo* with proteins showing high evolutionary correlation to it, in particular CLU, XCR1, GEN1, and IFNAR2. Similar binding affinity is predicted between CLU and GPR141 based on their high reciprocal rank ERCs. CLU-FGG and GPR141-FGB provide connections to fibrinogen based on their evolutionary correlations, suggesting binding affinities. Applicable methods could include protein complex immunoprecipitation, tagged protein analysis, and yeast-two-hybrid analysis (*Rao et al., 2014*).

We have begun preliminary analyses using short (10mer) amino acid sequences to identify predicted sites of interaction among protein partners. These data may be able to inform docking simulations for protein pairs using software that allows for the incorporation of *a priori* predicted interfaces (*Van Zundert et al., 2016*; *Pagadala, Syed &*

*Tuszynski, 2017*). For example, these 10mer analyses can be used to determine likely regions of binding affinity between ACE2 and Clusterin, for experimental validation through mutational analysis. Similarly, coagulation factor V shows high ERCs for non-canonical proteins, which can be investigated to determine whether F5 has novel functions outside of the coagulation pathway.

## CONCLUSIONS

In this paper, we take an exploratory approach to ACE2 protein interactions using evolutionary rate correlations. Our key findings are that the ERC analysis predicts ACE2 to have previously unidentified protein partners, and to be part of interaction networks relevant to COVID-19 pathologies. Most notably, ACE2 forms strong ERC networks relevant to coagulation and immunity. A potential mechanism is that reduced abundance of membrane-bound ACE2 disrupts these signaling networks. Additionally, the presence of the soluble ACE2 ectodomain may explain the systemic pathologies of COVID-19 infection as its circulation in the blood can affect pathways throughout the body.

We recognize that the new ACE2 protein connections predicted by ERCs may not be causal in severe COVID-19 pathologies. However, our novel findings that the ACE2 ERC network connects to coagulation and immunity pathways is noteworthy, with clear potential implications to some of the unusual features of COVID-19. In addition, results may have relevance to other functions of ACE2, such as circulatory homeostasis and digestion. The ERC analysis predicts additional protein connections that can be relevant to biological processes and disease. For instance, ERCs predict novel interactions for cytokine and immunity related proteins, such as for XCR1, IFNLR1, IFNAR2, and TLR8. Future investigations of the ERC networks of these and related proteins could be worthwhile. ERCs also suggest strong but previously undescribed connections for proteins such as CLU, GPR141, F5, and GEN1. Validation studies are necessary to determine to what extent strong ERCs predict biological interactions among proteins, such as the ones detected here.

Further computational analyses of ERCs are needed to better understand their relationship to protein function and evolution. For instance, machine learning and simulation approaches can be used to determine which aspects of protein structure, amino acid properties, and rates of protein evolution, improve ERC predictive power. We are currently expanding the mammalian protein set for such analyses. Finally, if evidence mounts that ERCs can be informative in predicting protein interactions, the approach can be applied more broadly as an additional tool for detecting protein interaction networks involved in many biological processes and disease.

## ACKNOWLEDGEMENTS

We thank Zhichao Yan for guidance with the mammalian protein database and discussions on ERC methods and independent contrasts. Also thanked are J. Fay, J. Fry, J. Jaenike, A. Kingsley, A. Larracuente, E. Sia, S. Ghaemmaghami for discussions and helpful feedback, and M. Tsuchiya for support and helpful discussions. Special thanks also go to Nathaniel and Helen Wisch for their support.

### Funding

This work was supported by a US National Science Foundation RAPID award to John H. Werren (2034507) and by the Nathaniel & Helen Wisch Chair Research Fund to John H. Werren. There was no additional external funding received for this study. The funders had no role in study design, data collection and analysis, decision to publish, or preparation of the manuscript.

### Grant Disclosures

The following grant information was disclosed by the authors:
US National Science Foundation RAPID: 2034507.
Nathaniel & Helen Wisch Chair Research Fund.

### Competing Interests

The authors declare that they have no competing interests.

### Author Contributions

- Austin A. Varela conceived and designed the experiments, performed the experiments, analyzed the data, prepared figures and/or tables, authored or reviewed drafts of the paper, and approved the final draft.
- Sammy Cheng performed the experiments, analyzed the data, authored or reviewed drafts of the paper, and approved the final draft.
- John H. Werren directed research conceived and designed the experiments, performed the experiments, analyzed the data, prepared figures and/or tables, authored or reviewed drafts of the paper, and approved the final draft.

### Data Availability

Additional text, figures, and tables are available in the Supplemental Files.

The ERC pipeline and additional supporting R code are available on GitHub: https://github.com/austinv11/ERC-Pipeline/.

Large data files, including the full ERC matrix, full enrichment results, trees used for ERC calculations, and complete results for additional tests on ERC results are available on figshare: Varela, Austin; Werren, John H. (2021): Supplemental Files. figshare. Dataset. DOI 10.6084/m9.figshare.14637450.v3.

### Supplemental Information

Supplemental information for this article can be found online at http://dx.doi.org/10.7717/peerj.12159#supplemental-information.

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
