# Peer review of "Novel ACE2 protein interactions relevant to COVID-19 predicted by evolutionary rate correlations"

_PeerJ, doi:10.7717/peerj.12159_

## Round 0.1 · original submission · Major Revisions

The reviewers were highly supportive of the work, though there were a number of questions, especially with regard to the interpretation of certain results and the size of included gene sets. Please try and address these questions as appropriate.

Reviewer 1 ·

Basic reporting

The authors present their study of evolutionary rates and their correlations between the ACE2 protein and ~2000 mammalian proteins. The evolutionary rate correlation (ERC) approach has been validated by them and other groups previously and has been used to identify new functional interactions in the past. The authors set their sites on the ACE2 protein which is the entry key for the SARS-CoV-2 virus. By examining the genes that correlate with ACE2's terminal branch rates, they find significant enrichments for coagulation cascade pathway proteins and interferon proteins. These represent potential new functional interactions for ACE2, which is known for catalyzing a chemical reaction of angiotensin and thereby affecting blood pressure.

The study appears to be conducted well and the findings are summarized in a set of tables and graphs. One concern to address is the relatively small proportion of the genome (~2,000 genes) that were surveyed and thereby potentially leaving out many important interactions. Second, the interpretation of ACE2's correlations as being strictly related to viral activity is too narrow and should consider other selective pressures that could be acting on ACE2. These and other points of potential concern are itemized below.

Generally, the study has significant contributions to make and appears to be well conducted. It is worthy of publication and would be of interest to many outside the evolutionary field. Major concerns were with interpretation, length, and small gene set.

Experimental design

1. Study would be strengthened by including more than 2,000 proteins. As is, it could likely be missing many key interactions. This low number also probably explains the low amount of overlap with the phylogenetic profiling study.

2. Reciprocal Ranks is an interesting idea, but the rationale behind it is speculative. Why not dig and find out why one gene ranks differently by examining their rate values and species representations? If all genes are constrained to the same set of species, does the rank discordance remain?

3. The idea of unidirectional networks is interesting but would need some support of the assertions they are making. Also, see alternative explanations above,

4. Line 303. It's not clear how the authors "sequentially extending branch lengths in different clades", so this analysis is hard to interpret.

5. Line 304. If the authors want the conclusion that rates increase as branch length to be a feature of their analysis, it would be necessary to determine if this is an artifact of phylogenetic methods. Simulate a set of genes under similar divergence times but with a uniform rate of change across branches and see if the same result is obtained.

6. Also, why not show any results for this (Lines 303+)? Plots, tables?

Validity of the findings

7. Viral pressures are surely not the only thing affecting ACE2's rates. ACE2 is the entry point for SARS-CoV2 but the observed correlations may not be related to the downstream pathologies induced by the virus. These associations are interesting but of less certain functional linkage to pathology which involves inflammation and cell death caused by cell lysis.
ACE2's general role, outside of being exploited by SARS-CoV2, is to regulate blood pressure through hydrolysis of angiotensin II, a vasoconstrictor peptide, into angiotensin, a vasodilator. Perhaps these general roles in modulating circulation are also reasons for the observed ERC correlations, rather than being linked to coronaviruses. Even new general, unappreciated functions of ACE2 could be interesting given its role in maintaining circulatory homeostasis, even if those functions were not related to viral activity.
Relevant to Line 502, "this is very relevant to COVID-19...", and in the framing of the entire study.

8. Relatedly, have coronaviruses being using ACE2 throughout mammalian evolution? That might be a requirement for the ERCs to be relevant to SARS-CoV2 pathology.

9. Could the authors demonstrate that characterized viral-associated proteins show high ERCs with their known interaction partners? To what degree are these accurate? Or outside of viruses, how well do ERC predict interactions for other cell surface proteins?
Before asking the reader to embark on interpreting ACE2's ERCs, it would help to show likelihood of these being correct.

10. Line 314, the authors state that ERC is predictive of direct protein interactions. However, previous studies by (Juan Pazos and Valencia. PNAS 2008) and (Clark Alani and Aquadro. Genome Research 2012) found strong ERC values for non-interacting proteins as well, such as those participating in metabolic pathways, so direct interaction does not seem to be not required.

11. Subnetwork analyses. Are the known functions of these genes significantly enriched within the networks? The worry is that it's always possible to find genes that "make sense" given a list of genes.

Additional comments

12. Large parts of Section C and C2 contribute relatively little to the manuscript and could be greatly truncated. The main findings were already presented in previous sections, and these repeat many of those points. If a journal needed a shorter format these could be shortened or left out.

13. These sections C and C2 are very descriptive and it is difficult to take away major conclusive points from them.

·

Basic reporting

no comment

Experimental design

The experimental design is robustly structured

Validity of the findings

The findings are very important as they suggest possible mechanisms for what is going on in vivo and suggested ideas for other interested people to investigate the coevolved proteins in practical namely the cytokine signaling pathway.

The authors pushed the idea of their article using different bioinformatics approaches to make robust conclusion with the support of what is published about proteins implicated in COVID-19 pathogenesis. So the quality of data is very convincing.

Additional comments

I think the following conclusion is based on weak evidences
“Thus, connections of FAM3D to F13B, possibly through 541 physical binding as implied by their high reciprocal ranks, could be important in modulating the 542 dynamics of clot formation”

I think the following conclusion is not clear needs to be rephrased:
“their protein 691 evolutionary correlations suggest that ACE2 may be part of a signaling network (perhaps 692 modulated through XCR1) that is contributory”

It will be great if you add a clarification regarding the correlation of ACE2 or any mediator with the plasma cell differentiation and antibody production such as Blimp-1, Predm1m and PAX5 as a try to understand the vaccination efficiency.

Do you have an idea about any coevolutionary correlation of ACE2 to IRF4 and IRF8? if you can make a clear conclusion about them please include it.

---

## Round 0.2 · accepted · Accept

Thank you for addressing the reviewer questions and concerns and congratulations again.

Reviewer 1 ·

Basic reporting

The authors have made clear the scope of their study in their responses. The analysis was not expanded to more genes, which in the end limits its impact. However, the results they provide based on analysis of 2000 genes are useful and could indicate new functional interactions of ACE2 with previously unexpected proteins.

Experimental design

I appreciate the textual edits and clarifications. It was also noted that the authors performed a new analysis on a gene pair to examine how species set representation affects reciprocal ranks.

Validity of the findings

My overall review is that the response was partial and declined to add analyses that would have strengthened the findings. For example, the authors still do not know if the new reciprocal ranking method is more informative than other potential ranking systems. However, the authors’ response argues that their ERC predictions are useful now and should be published as such. I agree and find the results to be sound. I recommend publication.

·

Basic reporting

No comment

Experimental design

I had no comments

Validity of the findings

Novel and helps to predict the correlation between covid 19 and post infection complications

Additional comments

The authors tackled all the comments that I was mentioned in the first reviewing process